# Paternal imprinting of *dosage-effect defective1* contributes to seed weight xenia in maize

Dawei Dai [1,10], Janaki S. Mudunkothge[1,10], Mary Galli [2], Si Nian Char [3], Ruth Davenport[4], Xiaojin Zhou[5], Jeffery L. Gustin [1,6], Gertraud Spielbauer[1], Junya Zhang[1], W. Brad Barbazuk[4], Bing Yang [3,7], Andrea Gallavotti [2,8] & A. Mark Settles [1,9] ✉

Historically, xenia effects were hypothesized to be unique genetic contributions of pollen to seed phenotype, but most examples represent standard complementation of Mendelian traits. We identified the imprinted *dosage-effect defective1* (*ded1*) locus in maize (*Zea mays*) as a paternal regulator of seed size and development. Hypomorphic alleles show a 5–10% seed weight reduction when *ded1* is transmitted through the male, while homozygous mutants are defective with a 70–90% seed weight reduction. *Ded1* encodes an R2R3-MYB transcription factor expressed specifically during early endosperm development with paternal allele bias. DED1 directly activates early endosperm genes and endosperm adjacent to scutellum cell layer genes, while directly repressing late grain-fill genes. These results demonstrate xenia as originally defined: Imprinting of *Ded1* causes the paternal allele to set the pace of endosperm development thereby influencing grain set and size.

The maize (*Zea mays*) endosperm constitutes the bulk of mature kernel weight. Endosperm develops from a diploid maternal central cell being fertilized by a haploid sperm cell with a predominant maternal genome influence on seed phenotype[1,2]. Prior to the rediscovery of Mendelian genetics, seed phenotypes caused by pollen genotype were hypothesized to be due to parental interactions analogous to xenia, the classical Greek guest-host relationship, in which the paternal genome is viewed as a guest with obligate roles for a successful interaction after fertilization[3]. Most xenia effects reveal dominant alleles of Mendelian traits, but parent-of-origin specific gene expression caused by genomic imprinting allows variation from individual parental alleles to determine seed phenotypes[4]. The first imprinted gene discovered was the maize *R*[r] allele[5]. Maternal bias in *R*[r] expression

causes mottled, reduced levels of kernel anthocyanins when *R*[r] is inherited through pollen.

More than 100 imprinted maize genes are known from allele-specific expression analysis[6,7]. Yet, only two maternally expressed genes (MEGs) of maize, in addition to *R*[r], have been shown to confer qualitative seed phenotypes, *maternally expressed gene1* (*meg1*) and *floury3* (*fl3*)[8–10]. No paternally expressed genes (PEGs) are known to have functional roles in kernel development, but three maize paternal-effect seed mutants have been reported[11].

Here, we show that the *dosage-effect defective1* (*ded1*) locus is a quantitative PEG that acts as a transcriptional regulator of endosperm developmental progression as well as promoting expression of genes critical for endosperm support of embryo development.

[1]Horticultural Sciences Department, University of Florida, Gainesville, FL 32611, USA. [2]Waksman Institute, Rutgers University, Piscataway, NJ 08854, USA. [3]Division of Plant Sciences, Bond Life Sciences Center, University of Missouri, Columbia, MO 65211, USA. [4]Department of Biology, University of Florida, Gainesville, FL 32611, USA. [5]Crop Functional Genome Research Center, Biotechnology Research Institute, Chinese Academy of Agricultural Sciences, Beijing, China. [6]United States Department of Agriculture, Urbana, IL 61801, USA. [7]Donald Danforth Plant Science Center, St. Louis, MO 63132, USA. [8]Department of Plant Biology, Rutgers University, New Brunswick, NJ 08901, USA. [9]Bioengineering Branch, NASA Ames Research Center, Moffett Field, CA 94035, USA. [10]These authors contributed equally: Dawei Dai, Janaki S. Mudunkothge. ✉e-mail: andrew.m.settles@nasa.gov

## Results

### *Ded1* is a seed weight dosage effect gene

We identified the *ded1-ref* allele from a quantitative screen of 1068 self-pollinated ears segregating for UniformMu defective kernel mutants[12]. Normal kernels from each ear were individually weighed and assayed for composition using a custom near infrared grain analyzer[13]. A cumulative distribution plot of *ded1-ref*/+ and +/+ kernel weights from this screen showed a 2 mg step increase between the lower 1/3 and upper 2/3 of kernel weights (Supplementary Fig. 1a). Although not significantly different from a normal distribution, the seed weight distribution had excess kurtosis of −0.72 indicating the lower and upper tails had greater than expected deviations from the mean. Kernels from the upper and lower 20% of seed weights were planted in separate cultures, self-pollinated, and scored for the *ded1* phenotype (Supplementary Fig. 1a). This seed weight sorting resulted in significantly different frequencies of *ded1* heterozygotes. Heavier kernels were more likely to be homozygous normal (Fisher's exact test, $p = 0.015$).

Homozygous mutants in W22, B73, or Mo17 genetic backgrounds typically produce a severe defective kernel with nearly an empty pericarp (Fig. 1a, b and Supplementary Fig. 1b, c). Ears segregating for *ded1* homozygotes show variable expressivity with 0–15% of mutant kernels being viable. Viable seeds develop into slightly smaller, fertile plants (Fig. 1c and Supplementary Fig. 1d). Self-pollination of *ded1-ref* plants produces all mutant seeds with accelerated anthocyanin accumulation in the aleurone (Fig. 1d, e).

Bulked segregant analysis and fine mapping of *ded1-ref* mutants from F$_2$ populations narrowed the *ded1* locus to 470 kbp on chromosome 1 encompassing 9 protein coding genes (Supplementary Fig. 2a, b). Genomic sequencing revealed a *copia*-like retrotransposon insertion in the R2R3-MYB transcription factor gene, *ZmMyb73* (Fig. 1f and Supplementary Fig. 2c). The predicted mRNA sequence for *ZmMyb73* varies among B73 genome annotations from version 2 to version 5. We amplified and sequenced the B73 *Ded1* endosperm cDNA, which confirmed the genomic transcriptional unit annotated in the B73_v4 gene model, Zm00001d033265. No evidence was found for a predicted alternatively spliced transcript in the B73_v5 gene model, Zm00001eb050770. PCR of *ded1-ref* mutant cDNA amplified a 5′ open reading frame (ORF) product but not a product 3′ of the retrotransposon insertion (Supplementary Fig. 2d). The full-length cDNA sequence of the *ded1-ref* transcript ORF includes part of the retrotransposon sequence and a predicted protein lacking the C-terminal acidic domain (Fig. 1f). Targeted mutagenesis of *ZmMyb73* with CRISPR-Cas9 produced four insertion-deletion alleles that segregated for kernel mutants and failed to complement *ded1-ref* (Fig. 1g and Supplementary Fig. 3).

The *ded1-1* to *ded1-4* alleles cause premature termination codons that truncate the *Ded1* ORF 5′ of *ded1-ref*. Quantitative RT-PCR (qRT-PCR) analysis of exon 1 in 12 days after pollination (DAP) endosperm revealed decreased expression of *ded1* in homozygous mutants for all alleles, which is consistent with the transcript being sensitive to nonsense mediated decay (Fig. 1h). Mutant *ded1-3* and *ded1-4* kernels are

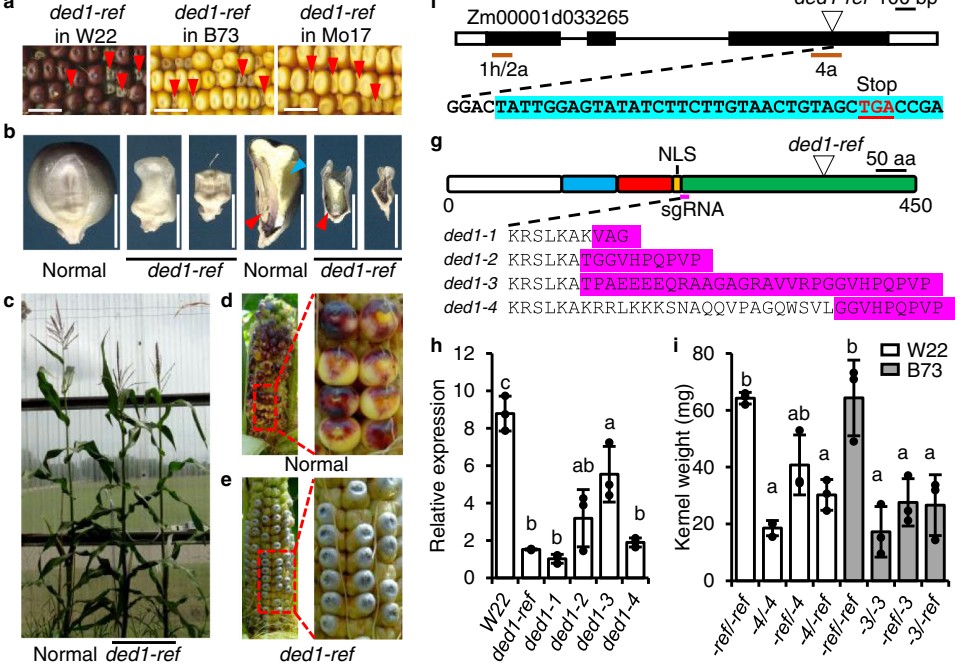

**Fig. 1 | *Ded1* is a transcription factor required for kernel development. a** Self-pollinated ears segregating for *ded1-ref* in W22, B73, and Mo17 genetic backgrounds. Arrowheads indicate mutants. Scale bars are 1 cm. **b** Kernel phenotype and sagittal sections of the *ded1-ref* and normal sibling in W22. Scale bars are 5 mm. Red arrowheads indicate the embryo, and blue arrowhead indicates vitreous endosperm. **c** Normal and *ded1-ref* sibling plants. **d** Self-pollinated normal sibling ear at 19 DAP. **e** Homozygous *ded1-ref* ear at 19 DAP. **f** Schematic of the B73_v4 genome annotation for the *Ded1* gene. Boxes are exons with coding sequences in black. Black lines are introns. Orange lines are the qRT-PCR products used in Figs. 1h, 2a, and 4a. The triangle indicates the *ded1-ref* retrotransposon insertion. The 5′ transposon junction sequence is highlighted in blue. **g** Schematic of DED1 protein domains showing R2 (blue) and R3 (red) MYB DNA binding domains, the nuclear localization signal (yellow), and C-terminal acidic domain (green). The

triangle indicates the *ded1-ref* insertion. Protein sequences of the Cas9-induced frameshifts are highlighted in fuchsia. **h** Endosperm expression of the *ded1* locus in W22 and homozygous *ded1* mutant alleles at 12 DAP. Relative qRT-PCR used 18 S rRNA as the control. Data are mean ± SD ($n = 3$ replicate PCR experiments from pooled mutant endosperm samples from a single ear). Letters denote significant differences ($P < 0.05$) from Tukey's HSD test. **i** Average kernel weight based on 50 homozygous mutant seeds from heterozygous *ded1-ref* (both W22 and B73 background), *ded1-3* (B73), and *ded1-4* (W22) self-pollinations and reciprocal crosses. Female parent is listed first. Mutant seeds from three ears were weighed for each genetic combination. Data shown as mean ± SD ($n = 3$ biologically independent ears from different plants with 50 mutant kernels sampled per ear). Letters denote significant differences ($P < 0.05$) from Tukey's HSD test.

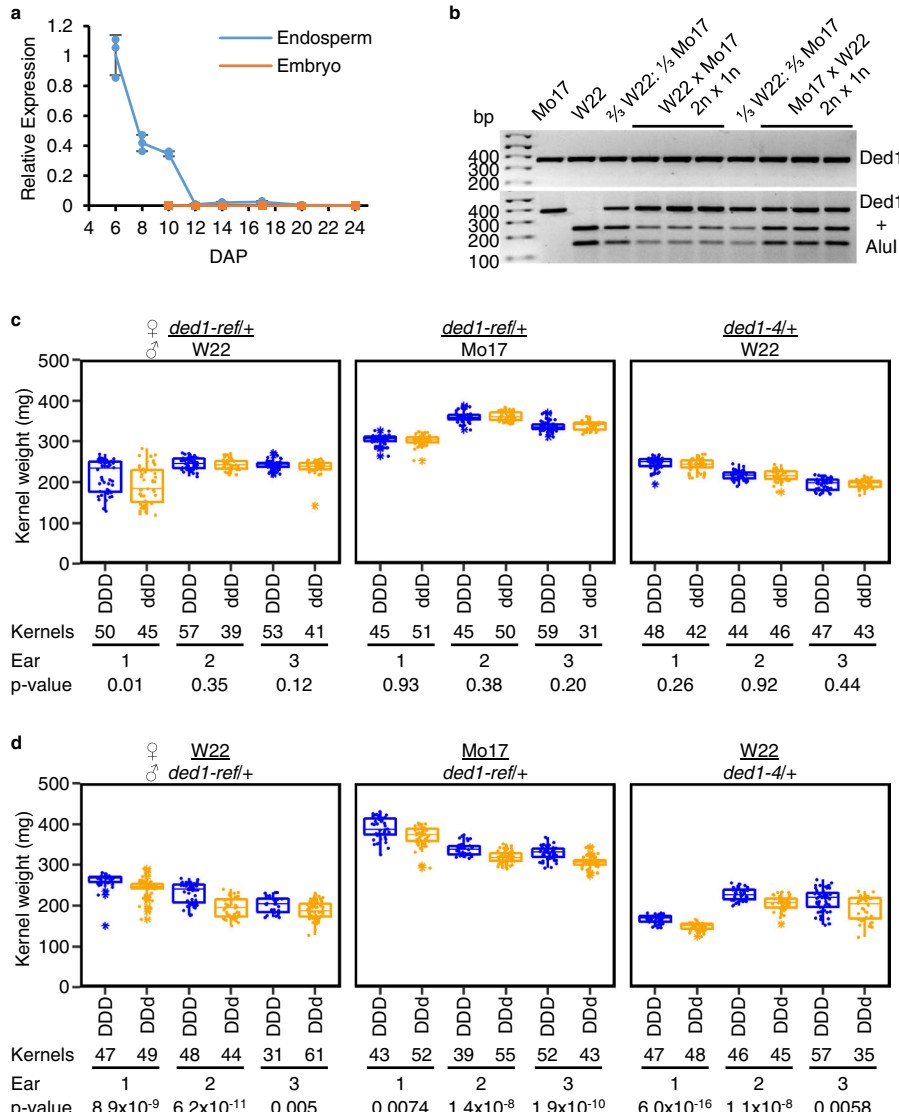

**Fig. 2 | Endosperm expression and paternal imprinting of *Ded1* affects kernel weight. a** Expression of *Ded1* in dissected W22 endosperm and embryo tissues. Transcript levels were normalized to 18 S rRNA. Data shown as mean ± SD ($n = 3$ replicate PCR experiments of pooled tissue samples from individual ears). **b** Allele-specific expression of *Ded1* in 12 DAP endosperm tissue comparing inbred self-pollinations and three biological replicates of reciprocal crosses between W22 and Mo17. The female parent is listed first with the endosperm genetic dosage below. Proportional mixes of cDNA derived from W22 and Mo17 inbred are indicated with fractional mix ratios. RT-PCR products were digested with *Alu*I to digest W22

products. **c, d** Box plots of homozygous normal and *ded1* heterozygous sibling kernels for maternal (**c**) and paternal (**d**) transmission of *ded1*. Boxes are the interquartile range with the median denoted by a horizontal line. Whiskers show the 1.5× interquartile range. Points show individual kernel weights with outlier kernels denoted with an asterisk. For each cross, the female parent is listed first. Homozygous normal (DDD) kernels are plotted in blue. Heterozygous kernels are plotted in orange. The *x*-axis also indicates the number of kernels plotted and *p* values for two-sided Student's *t* tests without correction for multiple tests. Individual kernel weights and descriptive statistics are in the Source Data file.

smaller than *ded1-ref* based on kernel weight and size (Fig. 1i and Supplementary Fig. 3c, e). Reciprocal crosses between *ded1-ref*/+ and *ded1-3*/+ or *ded1-4*/+ produce defective kernels with an intermediate seed weight suggesting that *ded1-ref* is a leaky hypomorphic allele.

### Quantitative imprinting of the *ded1* locus

Public transcriptome data show *Ded1* expression in endosperm from 5 to 13 DAP with a peak at 6 DAP[14–16] (Supplementary Fig. 4a). Specific expression in early endosperm was confirmed with qRT-PCR (Fig. 2a and Supplementary Fig. 4b). Moreover, *Ded1* is a quantitative PEG in endosperm transcriptome experiments of reciprocal B73 and Mo17 crosses[6,7,17]. Based on these studies, the paternal allele of *Ded1* shows stronger bias early in endosperm development and accounts for 75–77% of the total transcript at 10 DAP and 53–68% of total transcript at 14 DAP[6,7].

Genes can show allele-specific imprinting, and we tested the W22 allele of *Ded1* for imprinting in reciprocal hybrids with Mo17 using a cleaved amplified polymorphic sequence (CAPS) RT-PCR assay from 12 DAP endosperm RNA[18]. An *Alu*I restriction digest distinguishes W22 from Mo17 alleles (Fig. 2b). Mixes of inbred RNA emulated the expected relative expression levels of the inbred alleles based on two maternal and one paternal doses. Consistent with the prior RNA-seq analyses, the paternal allele in the reciprocal hybrids accounted for ~2/3 of total transcript amplified (Fig. 2b). We infer from the published transcriptome studies and the CAPS RT-PCR results that one copy of the paternal allele expresses at approximately fivefold greater level than one copy of the maternal allele. Consequently, loss of *Ded1* function is expected to have a greater impact on seed phenotype when inherited from pollen.

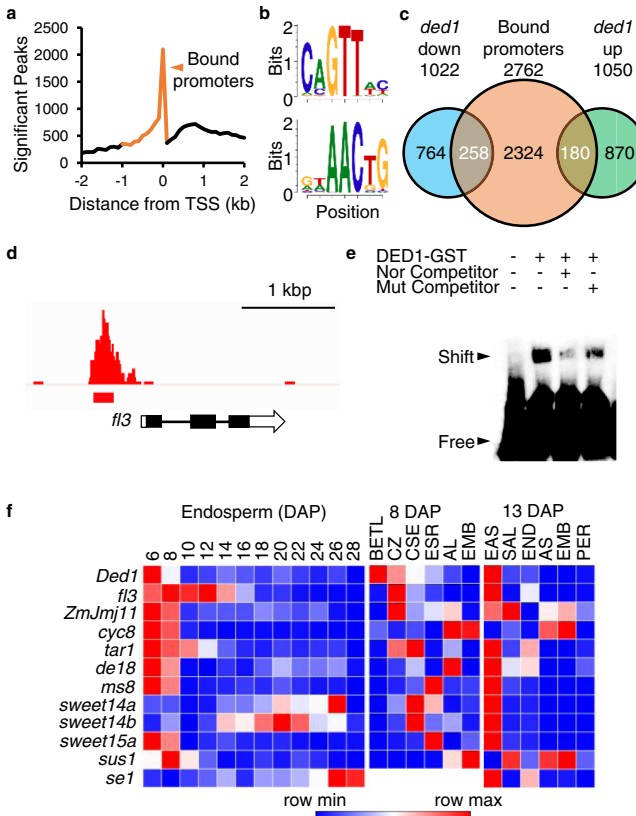

**Fig. 3 | Identification of DED1 target genes. a** Number of DED1 DAP-seq peaks per 100 bp bin relative to annotated transcriptional start sites. Orange line shows the bound promoter region used to identify direct target genes. **b** Sequence logo of top DED1 binding motifs identified by meme-chip v4.12.0 using the 5000 highest scoring DAP-seq peaks. **c** Venn diagram showing genes bound by DED1 from −1 to +0.1 kb of the TSS that were tested for differential expression (orange) as well as DEGs from 12 DAP *ded1-ref* and normal sibling endosperm ($q < 0.05$, FC > 2, and TPM > 1). Overlapping genes (white text) are inferred to be direct targets of DED1, including 258 DED1-activated and 180 DED1-repressed genes. **d** Genome browser visualization of DAP-seq reads at *fl3*. **e** EMSA with DED1-GST purified protein and labelled *fl3* promoter. Normal (Nor) or mutant (Mut) competitor probes were added at 100-fold greater concentration than the labelled *fl3* probe. This experiment was completed twice with similar results. **f** Public maize transcriptome data for *Ded1* and DED1 targets with known functions in endosperm development[15,25,26]. The color scale indicates the relative expression level of each gene. Expression of the *se1* locus was not detected in the 8 DAP dissected tissues.

Reciprocal crosses between heterozygous *ded1-ref*/+ or *ded1-4*/+ with inbred parents generated contrasting endosperm dosage levels of *ded1* mutant alleles (dd/D or DD/d) with homozygous normal siblings (DD/D) on the same ear. To evaluate the impact of genotype on kernel weights, 96 kernels from individual ears were sampled from the middle portion of the ear to avoid developmental position effects at the tip and base of the ear. When *ded1*/+ is used as female, the dd/D and DD/D genotypes have equivalent kernel weights in eight of nine biological replicates (Fig. 2c). By contrast, paternal transmission of *ded1* resulted in a 13–20 mg kernel weight reduction in the DD/d genotype depending upon the inbred background and allele (Fig. 2d). Combined with RT-PCR and published mRNA-seq results, *ded1* shows a seed weight reduction correlated with expression dosage. Paternal expression of one normal *Ded1* allele provides ~2/3 of normal transcript and complements maternally inherited *ded1* mutant alleles. Maternal expression of two normal *Ded1* alleles provides ~1/3 normal transcript and reduced seed weight is observed when *ded1* is inherited from the pollen.

## DED1 transcription factor targets

DED1 is a predicted transcription factor, and we confirmed it localizes to the nucleus by transient expression of a C-terminal fusion with enhanced green fluorescent protein (DED1-GFP) in maize protoplasts. DED1-GFP co-localizes with a red fluorescent protein (RFP) fusion with the nuclear localization signal (NLS) from SV40 large T-antigen (Supplementary Fig. 5a). We also tested DED1 for transcriptional activation using the yeast two-hybrid GAL4 DNA binding domain (BD) fusion. DED1, the ZmICEa transcriptional activator[19], and the non-activating LaminC proteins were expressed as translational fusions in the BD vector. Both DED1 and ZmICEa BD constructs expressed the *HIS3* and *ADE2* reporter genes without an activation domain construct (Supplementary Fig. 5b). Domain deletion analysis identified the C-terminal acidic domain as the activation domain in this assay (Supplementary Fig. 5b). The *ded1-ref* ORF also showed autoactivation, and the partial acidic domain from the DED1ref protein is sufficient for transcriptional activation. By contrast, DED1-ΔC, approximating the *ded1-3* or *ded1-4* predicted proteins, does not activate the reporter genes. These differences in transcriptional activation may explain the leaky phenotype of *ded1-ref*.

We investigated the molecular basis of DED1 developmental function by integrating differentially expressed genes (DEGs) from mutant-normal comparisons with DED1 DNA binding sites. First, DESeq2 analysis of endosperm transcriptomes from homozygous *ded1-ref* mutants and normal siblings at 12 DAP identified DEGs with at least a $\log_2$ twofold transcript level difference. There were 1022 DEGs with reduced expression in *ded1*, 1050 DEGs with increased expression (Supplementary Data 1). Gene Ontology (GO) term enrichment analysis found a diverse set of metabolic and response to stimuli terms in both increased and decreased DEGs (Supplementary Fig. 6a, b).

To identify DED1 DNA binding sites, a Halo-DED1 tagged protein was used to affinity purify B73 genomic fragments for DAP-seq[20]. A total of 43,225 DED1-enriched peaks were found with 61.6% of these binding sites within 1 kbp of the annotated transcriptional start sites (TSS) and stop sites of 15,367 annotated genes (Supplementary Fig. 6c). These DED1 binding sites are concentrated within 100 bp upstream of the TSS (Fig. 3a). Motif enrichment analysis of DED1 binding sites identified a similar binding motif to Arabidopsis MYB119[21] (Fig. 3b).

We focused on the 5860 genes with DAP-seq peaks −1 kbp to +100 bp of the TSS (Supplementary Data 2). This subset contained 2762 genes with detectable endosperm expression in the 12 DAP endosperm RNA-seq and included 438 DEGs (Supplementary Fig. 6d). The overlay of promoter binding and differential expression is a statistically significant enrichment over the DEG analysis alone (cumulative hypergeometric probability $= 2.2 \times 10^{-8}$) suggesting DEGs with DED1 binding sites are direct targets of DED1. There are 258 target genes with reduced expression in *ded1*, which can be inferred to be DED1-activated. A further 180 genes are DED1-repressed based on increased expression in *ded1* (Fig. 3c and Supplementary Data 3). Electrophoretic mobility shift assays (EMSA) with probes from DAP-seq peaks verified predicted binding of recombinant DED1-GST fusion protein to a subset of loci including *floury3* (*fl3*)[8], *sucrose synthase1* (*sus1*)[22], *colored aleurone1* (*c1*)[23], and *viviparous1* (*vp1*)[24] (Fig. 3d, e and Supplementary Fig. 6e, f). For each locus, DED1 binding was shown to be specific with non-labeled competitor probes. For *fl3* and *sus1*, we also replaced the DED1 binding motif of CAGTT with AGACC to generate unlabeled, mutant competitors. These mutant competitors did not interfere with DED1 binding activity as much as the unlabeled normal competitor indicating specific binding to the CAGTT motif (Fig. 3e and Supplementary Fig. 6f).

To better understand the developmental role of the *ded1* locus, we analyzed the expression pattern of *Ded1* and DED1 direct target genes using published RNA-seq studies. Specifically, we reanalyzed a

time course of endosperm development from 6 to 28 DAP as well as seed tissue dissections at 8 and 13 DAP[15,25,26]. Both *Ded1* and DED1-activated genes are enriched for peak expression from 6 to 8 DAP and are predominantly expressed in endosperm tissues of 8 DAP seeds (Fig. 3f, Supplementary Fig. 7a, and Supplementary Data 3). By 13 DAP, *Ded1* and 32% of DED1-activated genes have peak expression in the endosperm adjacent to scutellum (EAS) cell layer that is adjacent to the embryo. The EAS transcriptome is enriched for metabolite transporters that are hypothesized to direct nutrition from the endosperm to the embryo.

By contrast, DED1-repressed genes are enriched for peak endosperm expression post 10 DAP (Supplementary Fig. 7b). At 8 DAP, 68% of the repressed targets show peak expression in the embryo or aleurone (Supplementary Data 3). By 13 DAP, 80% of DED1-repressed targets have peak expression in the embryo, pericarp, and scutellar aleurone. Importantly, expression of direct targets correlates with the *Ded1* expression pattern from 8 and 13 DAP tissue dissections. At 8 DAP, *Ded1* is specific to endosperm with a lower level of expression only in aleurone, and by 13 DAP, *Ded1* expression is specific to EAS (Fig. 3f).

### Developmental mechanism of *ded1* dosage and xenia effects

Histology of homozygous *ded1-ref* and normal sibling seeds show that the *ded1* mutant endosperm fails to support embryo development. At 12 DAP, normal siblings have a fully developed body plan with embryonic roots and leaves (Supplementary Fig. 8a). By contrast, the *ded1* embryo is typically arrested at the pre-transition stage (Supplementary Fig. 8b). In addition, the *ded1* mutants have an incompletely developed basal endosperm transfer layer (BETL), which is critical for nutrient uptake from maternal tissues (Supplementary Fig. 8c, d).

Some DED1 direct targets have well-defined biological functions that give insight into the developmental consequences of *ded1* mutations (Supplementary Fig. 3f). Transcription factors comprise 7% (32/438) of direct targets with 18 genes activated and 14 genes repressed (Supplementary Data 3). Among activated targets, *floury3* (*fl3*), a PLATZ domain transcription factor causes an opaque endosperm and reduced grain-fill phenotype when mutated[8]. Among DED1-repressed targets, both *c1* and *pl1* are MYB transcription factors that regulate anthocyanin biosynthesis[23,27,28].

Other DED1-activated targets are associated with endosperm developmental processes. For example, *ZmJmj11* encodes a H3K27me3 demethylase that balances the activity of the Polycomb Repressor Complex2 repressive chromatin mark associated with imprinted gene expression[29,30]. The *cyclin8* (*cyc8*) locus encodes a B2-type cyclin associated with early endosperm cell proliferation[31]. The *tryptophan aminotransferase related1* (*tar1*) gene synthesizes the hormone, auxin[32]. The *defective endosperm18* (*de18*) locus encodes a YUCCA-related auxin synthesis enzyme and is potentially directly activated by DED1[33]. In *ded1* mutants, *de18* expression is significantly reduced, the *de18* locus has a DAP-seq peak 1.8 kbp upstream the TSS (Supplementary Fig. 6e).

Several DED1 direct targets have predicted functions in the endosperm-embryo interface. The *male sterile8* (*ms8*) locus supports pollen development and encodes a putative β−1,3-galactosyltransferase associated with secretory cell types in anthers[34,35]. In endosperm, *ms8* is specifically expressed in the embryo surrounding region (ESR) and EAS (Fig. 3f). Predicted transporters are 6% (28/438) of DED1 targets (Supplementary Data 3). Sugar transporters including *sweet14a*, *sweet14b*, and *sweet15a* are likely required for sugar uptake during seed development as the *sweet4c* locus has been shown to be essential for grain-fill[36]. Although *sweet14a* and *sweet14b*, show later expression during endosperm development, the genes have specific expression in the EAS (Fig. 3f).

There are also direct targets that function in the utilization of sucrose during seed development. DED1 directly activates *sucrose*

*synthase1* (*sus1*), a critical enzyme for nutrient uptake and starch biosynthesis[37]. DED1 also represses *sugary enhancer1* (*se1*), which is a repressor of starch catabolic genes that acts late in endosperm development to promote starch accumulation in *sugary1* mutants[38]. The range of molecular functions for genes directly regulated by DED1 demonstrates that *ded1* is a global regulator that promotes early seed development as well as endosperm-embryo interactions, while it represses later grain-fill functions.

Intriguingly, DED1 direct targets include imprinted genes (Supplementary Data 3). There are six MEGs that are evenly split between being activated or repressed by DED1. Among PEGs, 8/9 genes are DED1-activated targets including the *tar1* auxin biosynthesis locus. The *de18* locus is an additional PEG involved in auxin biosynthesis that may be directly activated by DED1.

As a transcription factor, the seed weight dosage effects observed for the *ded1* locus are expected to be mediated through changes in endosperm transcript levels. We generated a dosage series of 12 DAP endosperm RNA samples with controlled pollinations of *ded1-ref*/+ and W22 plants. Self-pollination of *ded1-ref*/+ produced homozygous mutant dd/d endosperms, and self-pollination of W22 produced DD/D (normal expression of *Ded1*). Reciprocal crosses generated dd/D (2/3 normal expression expected) and DD/d (1/3 normal expression expected) endosperms that were identified by genotyping dissected embryos. The dosage series was then analyzed by qRT-PCR for expression of the normal *Ded1* allele and six DEGs (Fig. 4a). To assay only the normal *Ded1* allele transcript level, primers were designed to amplify across the LTR insertion site in *ded1-ref* (Fig. 1f). These primers failed to detect *Ded1* transcript in the ddd genotype and showed parent-of-origin expression levels. Paternal inheritance in the ddD genotype had 50% *Ded1* transcript levels of the normal DDD genotype. When *Ded1* was maternally inherited in the DDd genotype, the normal allele transcript was at 15% levels of the DDD genotype. These data are consistent with the PEG pattern of expression observed in mRNA-seq studies and with the CAPS RT-PCR analysis in Fig. 2b.

The DED1-activated target gene, *fl3*, shows a threshold response to increased expression of the normal *Ded1* allele. There is only a 20-fold increase of *fl3* expression in the DDd genotype over the ddd homozygous mutant genotype, while the ddD heterozygote has 133-fold increase over the ddd mutant. A similar threshold effect was observed for the potential direct target, *de18*, where the DDd and ddD genotypes express at 32- and 149-fold greater than ddd, respectively.

DEGs without a DAP-seq binding peak also show expression level changes based on *ded1* dosage but with distinct patterns. The *sweet4c* hexose transporter has a threshold effect with the ddd homozygous genotype having a ~3-fold reduction of expression relative to other genotypes. The *tcrr1* response regulator shows a progressive decrease of expression correlated with *ded1* expression. We also analyzed two indirectly repressed genes, *22 kDa alpha zein5* (*az22z5*) and *anthocyaninless2* (*a2*). Both showed transcript level increases in the homozygous mutant ddd genotype. We conclude that paternal transmission of *ded1-ref* results in transcript level changes for a plurality of direct and indirect DED1 targets. The transcript changes from paternal transmission of *ded1-ref* are likely to mediate the seed weight reduction observed in ears segregating for DDD and DDd doses.

## Discussion

The parental conflict or kinship theory for the origin of imprinting in angiosperms and mammals predicts that MEGs and PEGs are selected to improve the fecundity of the maternal and paternal genomes, respectively[39,40]. Although there are many examples of angiosperm MEGs with essential roles in seed development[41], *ded1* is the only PEG of which we are aware that regulates seed development directly. Hundreds of PEGs have been identified in multiple plant species[6,7,42–48]. However, mutants for a subset of Arabidopsis PEGs produced no obvious effect on

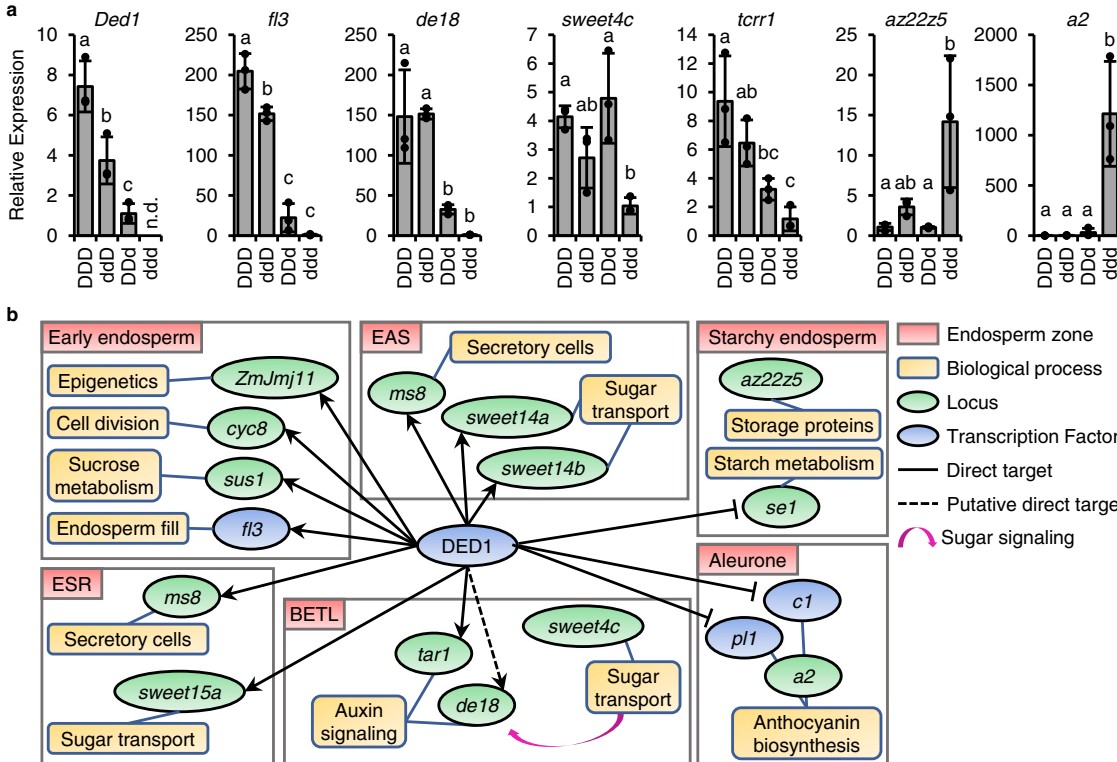

**Fig. 4 | Dosage-dependent regulation of downstream DED1 targets. a** Expression of the normal allele of *Ded1* and DED1 downstream genes by qRT-PCR in four dosage states of *Ded1* (D) and *ded1-ref* (d). Genes analyzed included the *fl3* direct target, *de18* potential direct target, *sweet4c* activated DEG, *tcrr1* imprinted DEG, *az22z5* α-zein repressed DEG, and *a2* repressed DEG. Data points are averages of three technical replicates. Bars and error bars are the mean ± SD of three biological replicates from independent ears (*n* = 3). Letters denote significant differences (*P* < 0.05) from Tukey's HSD test. **b** Schematic of DED1 target and downstream loci with documented roles in endosperm development. DED1 activates loci that act early in endosperm development and in nutrient transfer tissues. DED1 represses loci acting later during grain fill.

---

the seed[49,50]. Thus, *ded1* provides a critical example in relationship to parental conflict theory[51]. A paternally inherited *Ded1* allele is sufficient to promote embryo development and normal seed weight. Maternally inherited expression of *Ded1* is sufficient for seed development but with a reduced weight. The *Ded1* parent-of-origin expression pattern and seed weight outcomes illustrate an example where paternal inheritance of functional *Ded1* increases nutritional resource uptake and seed reserve accumulation.

At a molecular level, DED1 is a transcription factor that regulates early endosperm gene expression in a dosage sensitive manner. Consequently, the study has limitations in identifying DEGs and direct target genes. DEGs were identified from 12 DAP endosperm tissues that compared homozygous mutant to a pool of normal tissues with three doses of the normal *Ded1* allele. These mixed endosperm genotypes in the normal RNA extraction as well as sampling after 6 DAP, when *Ded1* is maximally expressed, likely reduced the statistical power to detect DEGs. Moreover, steady-state transcript levels combine both direct transcriptional responses to DED1 protein and feedback mechanisms that cause indirect transcript level changes.

As a quantitative PEG, *Ded1* is also expressed from maternally inherited alleles, and only homozygous mutant *ded1* seeds abort development. Inheritance of functional *Ded1* alleles solely from the maternal parent perturbs a subset of activated and repressed genes even when a gene is indirectly regulated through *ded1*. By contrast, paternal expression of *Ded1* generally is sufficient to achieve near normal steady-state transcript levels for both direct and indirect target genes. At a minimum, paternal *Ded1* is required for full seed size, and it becomes essential if the maternal allele is defective. Consequently,

*ded1* is an example of the xenia paternal guest-gift envisioned by Wilhelm Focke in 1881.

The expression patterns both of *Ded1* and its direct targets suggest developmental mechanisms regulated by DED1 (Fig. 4b). Early endosperm undergoes rapid nuclear divisions in a syncytium followed by cellularization and additional cell proliferation[52]. From 6 to 12 DAP, endosperm cell types are specified and differentiate. *Ded1* and many DED1-activated targets show peak expression 6–8 DAP with predicted functions in endosperm proliferation, such as *cyc8*, and cell type differentiation, such as *fl3* and *ms8*. Late endosperm differentiation genes, such as *c1*, *pl1*, and *se1*, are expressed precociously in *ded1* mutants supporting a role to delay differentiation associated with grain-fill. A PEG like *Ded1* that promotes endosperm growth is a direct prediction of parental conflict theory as proposed by Haig and Westoby[51].

At 13 DAP, *Ded1* and many activated targets are restricted to the EAS suggesting a role at the interface between endosperm and embryo. Even though *Ded1* is not expressed in embryos and *ded1* mutant embryo-escapes can complete a full life cycle, most *ded1* mutant embryos arrest at pre-transition stage. Based on prior studies, we speculate that *ded1* defects in ESR or EAS cells disrupt embryo development, while BETL defects are less likely to cause embryo arrest. BETL developmental defects can result from multiple indirect causes such as reduced seed sink strength as discussed in ref. 53. In addition, genetic variants in *meg1* or *sweet4c* that cause primary defects in BETL development do not result in embryo development defects[9,36]. Activated DED1 direct targets in the ESR and EAS, including *sweet14a*, *sweet14b*, and *sweet15a*, support a direct role in promoting endosperm transport of nutrients to the developing embryo. By contrast,

BETL-specific *sweet4c* does not have DED1 binding sites, and *sweet4c* expression is inferred to be reduced in *ded1-ref* by indirect mechanisms.

Our data support a model in which DED1 is a central regulator of seed development, but angiosperms show large variation in endosperm developmental progression. The low conservation of imprinted gene expression among angiosperms makes it likely that DED1 may only show xenia effects within cereals. A protein sequence alignment of DED1 homologs in Arabidopsis and cereal species reveals conservation primarily in the DNA-BD with a cereal-specific C-terminal domain (Supplementary Fig. 9). The most closely related Arabidopsis gene is MYB119 and has an essential developmental function in the female gametophyte[54]. The Arabidopsis and maize DED1 homologs have diverged, but the transcription factor lineage has maintained an essential function in reproductive development.

No Arabidopsis PEGs have been identified that impact seed development in diploid crosses. However, several PEG loci function in blocking 2n × 4n crosses[49]. In addition, the redundant MADS-box transcription factor loci, *phe1* and *phe2*, can also rescue 2n × 4n seed development[55]. Although *phe2* is biparentally expressed, *phe1* is a PEG, and the double mutant suppresses interploidy seed abortion when inherited through pollen. PHE1 target genes have overlapping developmental functions with DED1 targets such as a bias for regulating PEGs, epigenetic regulators, auxin biosynthesis genes, and non-imprinted transcription factors. Potentially, other PEGs within Arabidopsis have xenia functions analogous to *ded1*.

In summary, our work identified a major transcriptional regulator of seed size, which plays a key role during endosperm cell proliferation, endosperm differentiation, and promotes embryo development. Imprinted gene expression of *Ded1* confers a paternal control over normal development and resource allocation to the seed. These results support the kinship theory for the evolution of imprinted gene expression and the existence of xenia relationships within angiosperm reproductive development.

## Methods

### Genetic stocks
Maize was grown in the field at the University of Florida Plant Science Research and Education Unit in Citra, FL or in a greenhouse at the Horticultural Sciences Department in Gainesville, FL. The *ded1-ref* allele was identified in a quantitative screen of 1068 independent seed mutant isolates from the UniformMu population[12]. For each mutant, approximately 60 normal kernels were selected from $M_3$ segregating ears. Individual kernels were weighed and a single-kernel near infrared reflectance spectrum was collected using a custom grain analyzer[13]. Kernel weights for each $M_3$ ear were screened for distinct weight clusters by three researchers using cumulative distribution plots presented in a random order. When an ear was scored as having multiple kernel weight classes by 2/3 researchers, 96 normal kernels from the putative dosage-effect mutant were weighed and indexed to select the heaviest 20 kernels and lightest 20 kernels. These kernel selections were planted in separate cultures, self-pollinated, and crossed to color-converted W22 inbred plants. Self-pollinations of this $M_4$ generation were scored for defective kernel phenotypes, and the frequency of $M_4$ heterozygotes in the heavy and light kernel selections was tested for distortion using Fisher's Exact Test. Heterozygous *ded1-ref* plants were crossed to B73 and Mo17. The $F_1$ crosses were used to generate $F_2$ populations and to backcross *ded1-ref* into B73 and Mo17.

Genome-edited *ded1* alleles were isolated from crosses using ten independent events of the CRISPR/Cas9 editing construct transformed into the HiII genotype using *Agrobacterium tumefaciens* at the Iowa State University Plant Transformation Facility (www.biotech.iastate.edu/ptf/). $T_0$ plants were self-pollinated and crossed to W22, B73, and Mo17 inbred plants. $F_1$ transgenic plants were selected using 2% glufosinate-ammonium, self-pollinated to

screen for defective kernel phenotypes, and backcrossed to the recurrent inbred parent. The *ded1* locus was amplified and sequenced from individual lines segregating for seed phenotypes. Non-transgenic segregants of four Cas9-induced insertion-deletion alleles of *ded1* were selected for complementation tests with *ded1-ref* and further introgression into inbred lines.

### Molecular cloning of *ded1-ref*
The *ded1-ref* allele was mapped with bulked segregant analysis using a pool of 100 homozygous mutant kernels from the B73 × *ded1-ref*/+ $F_2$ mapping population. DNA was extracted from the pooled kernels and genotyped with 129 single nucleotide polymorphism markers using the Sequenom MassARRAY platform at the Iowa State University Genomic Technologies Facility as described previously[56]. Enrichment for the W22 allele at each marker was calculated as a ratio of ratios where the numerator was the ratio of W22/B73 signals for the BSA sample and the denominator was the ratio of W22 and B73 signals from non-segregating pooled samples. To reduce noise in the ratio calculations, all denominator values less than one were replaced with a value of one. The map position was identified by visual inspection of the W22 enrichment plot. Marker locations for the plots were taken from the Intermated B73xMo17 genetic map (https://www.maizegdb.org/data_center/map). The chromosome 1 map position was confirmed with the simple sequence repeat markers: phi037, umc1955, and bnlg1671. Fine mapping was completed by genotyping genomic DNA from 362 $F_2$ and 70 $BC_1S_1$ individual mutant kernels from Mo17 × *ded1-ref*/+ crosses using seven InDel markers indicated in Supplementary Fig. 2 and Supplementary Table 1.

The *ded1-ref* allele polymorphism was discovered by amplifying genomic fragments from *ded1* mutants within the fine map interval. The W22 *Ded1* allele was amplified with the primers Ded-30-F5 and MYB73-RT-R4. The *ded1-ref* allele was amplified with MYB73-F6-A and MYB73-RT-R4 primers. The PCR products were cloned into TOPO Blunt DNA cloning vector (Invitrogen) and sequenced.

Additional mutant alleles of *ded1* were generated by cloning the synthetic guide RNA sequences: 5′-GCGCTGATCAAGGCGCA-CAAGCGG-3′ and 5′-ACCACTGGAACGCCACCAAGCGG-3′ into the ISU Maize CRISPR cloning and transformation system as described[57]. Ten transformed callus lines were used to regenerate 65 plantlets that were propagated in the greenhouse via self-pollinations and crosses onto B73 inbred or *ded1-ref*/+ plants. CRISPR lines that failed to complement *ded1-ref*/+ were selected for further analysis. Non-transgenic segregants of CRISPR-induced *ded1* alleles were identified from $T_2$ or $BC_1S_1$ plants via sensitivity to glufosinate ammonium and complementation tests with *ded1-ref*/+ plants. DNA from homozygous mutant seeds was used to amplify and sequence the *ded1* locus to identify four insertion-deletion alleles for analysis.

### Expression analysis by RT-PCR
RNA was extracted as described[58]. Briefly, 250 mg of frozen, ground tissue was mixed with 200 µL of RNA extraction buffer (50 mM Tris-HCl, pH 8, 150 mM LiCl, 5 mM EDTA, 1% SDS in DEPC treated water). The slurry was extracted using phenol:chloroform and Trizol (Invitrogen). RNA was precipitated from the aqueous fraction using isopropanol and washed with 70% ethanol. RNA pellets were re-suspended in nuclease free water (Sigma) treated with Purelink DNase (Invitrogen). RNA was further purified using an RNeasy MinElute Cleanup Kit (Qiagen). DNA was removed by treatment with PureLink DNase (Invitrogen). RNA was reverse transcribed with M-MLV reverse transcriptase and Oligo(dT) primer (Promega). When applicable, PCR products were cloned into the TOPO Blunt DNA Cloning vector (Invitrogen) and sequenced.

The *Ded1* gene model was confirmed by sequencing RT-PCR products from 12 DAP W22 endosperm cDNA that was amplified with

MYB73-F15 and MYB73-RT-R4 primers. The 3′ end of the *ded1-ref* mutant transcript was amplified using the 3′-full RACE core set (Takara Bio) according to the manufacturer's instructions. Briefly, 1 μg of total RNA extracted from 12 DAP W22 and *ded1-ref* homozygous mutant endosperms were reverse transcribed with the oligo dt-3′ site adaptor primer and diluted cDNA (1:50) was PCR amplified with MYB73-F6-A and 3′ site adaptor primers.

Quantitative RT-PCR (qRT-PCR) was performed using a StepOnePlus real-time PCR machine (Applied Biosystems) with 1×SYBR Green PCR Master Mix (Applied Biosystems). Normalization was relative to 18 S rRNA using the comparative cycle threshold (ΔΔCt) method[59]. Primers are listed in Supplementary Table 1.

Allele-specific expression of W22 and Mo17 *Ded1* alleles was detected with a CAPS marker based on an *Alu*I restriction site unique to the W22 allele. *Ded1* cDNA was amplified with MYB73-CAPS-L1 and MYB73-CAPS-R1 primers and 25 μL PCR products were digested with 20 U *Alu*I at 37 °C for 4 h prior to gel electrophoresis.

### Individual kernel weight dosage analysis

Reciprocal crosses between W22 and *ded1-ref*/+, Mo17 and *ded1-ref*/+, and W22 and *ded1-4*/+ generated ears segregating 1:1 for homozygous normal and *ded1* heterozygotes. For each ear, 96 kernels were sampled from the middle region to avoid the tip and base of the ear. The kernels were weighed using the single-kernel grain analyzer and indexed in 48-well microtiter plates. The *ded1-ref* allele was genotyped using primers MYB73-F6-A, MYB73-R5, and retrotransposon-specific LTR-F2. The *ded1-4* allele was genotyped using a CAPS marker with primers DED1-30-F5 and MYB73-R9 followed by a restriction enzyme digestion with *Sty*I.

### Subcellular localization

The ORFs of GFP and DED1 were amplified using primers GFP-6, GFP-7, 73ORF-F4, and 73ORF-R4. The resulting fragments were In-Fusion (Takara Bio) cloned into the BamHI/BstEII and BamHI sites of pUB-iGUSPlus (Addgene), respectively, to generate pUbiGFP-DED1. For a nuclear-localized marker, the SV40 (MPKKKRKV) NLS was fused to the N-terminal of RFP by PCR amplification using primer set RFP-F1 and RFP-R2. The fusion fragment of *NLS-RFP* was amplified using primers RFP-F2 and RFP-R2 and cloned into the BamHI/BstEII site of pUB-iGUSPlus to generate pUbiNLS-RFP. The GFP-DED1 and NLS-RFP were transiently expressed in maize mesophyll protoplasts as previously described[60]. Briefly, maize seedlings were grown in a growth chamber in the dark for 7–10 d. Leaf blade tissue was cut into strips and digested in 1.5% (w/v) cellulose R10, 0.4% (w/v) macerozyme R10, 20 mM MES (pH 5.7), 0.4 M mannitol, 20 mM KCl, and 0.1% (w/v) BSA in the dark at 28 °C for 3–4 h. Released protoplasts were washed in 2 mM MES (pH 5.7), 154 mM NaCl, 125 mM CaCl₂, and 5 mM KCl, and then resuspended in 4 mM MES (pH 5.7), 0.4 M mannitol, and 15 mM MgCl₂. The protoplasts were transfected in 40% (w/v) PEG-4000, 0.2 M mannitol, 100 mM CaCl₂. Fluorescence was imaged using spinning disc confocal microscopy (X81-DSU; Olympus) 12–16 h after transient transformation.

### Yeast transactivation assay

Transactivation was tested using the autoactivation assay from the Matchmaker Gold Yeast Two-Hybrid System (Clontech). ORFs for DED1 and DED1 truncations were amplified from the DED1 cDNA using combinations of three different forward (DED1-BD-F1 to F3) and four reverse (DED1-BD-R1 to R4) primers depicted in Supplementary Fig. 5b and listed in Supplementary Table 1. Each product was subcloned into the yeast expression plasmid pGBKT7 with the GAL4 DNA BD. The ZmICEa ORF was amplified as an RT-PCR product from 12 DAP B73 seed RNA and cloned in pGBKT7. Autoactivation was tested in the yeast strain AH109 and were scored based on the growth of transformed yeast on SD/-Trp -His -Ade plates.

### Identification of direct targets of DED1

For RNA-seq, normal and *ded1-ref* sibling 12 DAP endosperm tissues were dissected from four self-pollinated plants and frozen in liquid nitrogen. Total RNA was extracted from each biological replicate of paired *ded1-ref* and normal sibling samples. Strand-specific TruSeq (Illumina) cDNA libraries were prepared and raw RNA-seq reads were processed as described[58]. Briefly, 1 μg of total RNA was input for non-strand-specific TruSeq (Illumina) cDNA libraries with a median insert length of 200 bp. Raw RNA-seq data were screened to remove adapter sequences using Cutadapt v1.1 with the following parameters: error rate = 0.1, times = 1, overlap = 5, and minimum length = 0. Adapter trimmed sequences were quality trimmed with Trimmomatic v0.22 using parameters (HEADCROP:0, LEADING:3, TRAILING:3, SLIDINGWINDOW:4:15, and MINLEN:40) to truncate reads for base quality <15 within 4 base windows and kept only reads ≥40 bases after trimming. Reads were uniquely aligned to the B73 RefGen_v3 maize genome assembly with GSNAP (Version 2013-07-20) using the following parameters: orientation = FR, batch = 5, suboptimal levels = 0, novel splicing = 1, local-splice dist = 8000, local-splice penalty = 0, distant-splice penalty = 4, quality protocol = sanger, npaths = 1, quiet-if-excessive–max-mismatches = 0.02, no fails-format = sam, sam-multiple-primaries–pairmax-rna = 8000, pair expect = 200, pair dev = 150, nthreads = 4. Differentially expressed transcripts were detected with the DESeq2 Bioconductor package[61]. Transcripts were considered expressed if >1 transcript per million was detected in at least one genotype, and a total of 16,219 gene models met this criterion. DEGs were required to have an adjusted $p < 0.05$ and a $Log_2$(Fold Change)>2. GO-term enrichment was analyzed using agriGO v2.0 with default parameters[62].

For DAP-seq, maize B73 genomic DNA adapter ligated libraries were prepared according to the established protocol[20]. Briefly, 5 μg phenol:chloroform:isoamyl alcohol prepared genomic DNA from maize developing B73 ear (~1 cm) was diluted in EB (10 mM Tris-HCl, pH 8.5) and sonicated to 200-bp fragments using a Covaris S2 sonicator. DNA was purified using AmpureXP beads at a 2:1 bead to DNA ratio. Samples were end-repaired using the End-It kit (Lucigen), processed with a Qiaquick PCR purification (Qiagen) kit, and then A-tailed using Klenow fragment (3′→5′exo-) for 30 min at room temperature. The samples were processed with Qiaquick PCR purification kits and then ligated overnight with a truncated Illumina Y-adapter as described previously[20]. Libraries were purified by bead cleaning using a 1:1 bead:DNA ratio, eluted from the beads in 30 μL of EB, and quantified with the Qubit HS fluorometric assay.

The DED1 ORF was cloned into the pIX-HALO plasmid and HALO-DED1 fusion protein was expressed using the TNT rabbit reticulocyte expression system (Promega)[20]. As a negative control, HALO-GST was expressed in parallel. Protein purification and the DAP-seq assay were performed as described previously[63]. Briefly, 1 μg of pIX-HALO-DED1 or HALO-GST plasmid DNA was expressed in a 50 μL rabbit reticulocyte TNT expression reaction for 2 h at 30 °C. The TNT reaction was added to 10 μL of washed Magne-HALO beads (Promega) and 40 μL of 1× Phosphate Buffered Saline with 0.005% NP-40 (1xPBS + NP40), and mixed with a tube rotator at room temperature for 1 h. Protein-bound beads were subsequently washed four times with 1xPBS + NP40 and re-suspended in 100 μL of 1×PBS + NP40 containing 1 μg of maize B73 adapter ligated library. Samples were mixed with a tube rotator at room temperature for 1 h. Samples were then washed 8–10 times in 1×PBS + NP40. The beads were re-suspended in 30 μL of EB after the final wash, incubated at 98 °C for 10 min, and placed on ice. The samples were placed on a magnet to separate the beads, and the DNA-containing solution was removed by pipetting. DNA enrichment and index adapters were added prior to 75 bp single-read sequencing on an Illumina NextSeq500.

DAP-seq reads were trimmed[64] and mapped to the B73_v3 reference genome using bowtie2 v2.2.853[65] with default parameters.

Uniquely-mapped reads were kept for further analysis. Peaks were called using GEM v2.5[66] using a Benjamini-Hochberg adjusted $p$ value ($q$ value) threshold of $q = 0.00001$ (option−q 5), while excluding a list of common false positive sites from Galli et al.[63]. Additional background subtraction used sites identified in the HALO-GST in vitro expressed protein sample. Motif enrichment analysis was completed with GEM using parameters−k_min 6−k_max 20. Motif logos were generated using MotifStack[67]. Bigwig files were used to visualize the peaks in the Integrative Genomics Viewer[68].

Public maize endosperm transcriptome analysis was downloaded from published supplementary files[15,25,26]. FPKM transcript levels were transformed to transcripts per million (TPM). Direct target genes were clustered using the Morpheus tool (https://software.broadinstitute.org/morpheus) with hierarchical clustering and parameters of: metric = "one minus pearson correlation", linkage method = "complete", and clustering by gene "rows". Heat maps were visualized with a relative color scheme for each gene to illustrate peak expression for the individual gene.

## EMSA
The *Ded1* ORF was cloned into pGEX-4T-1 (Clontech) to express GST-DED1 fusion protein in *E. coli* strain BL21. The fusion protein was purified using the MagneGST Protein Purification System (Promega). Oligonucleotide probes containing DED1 predicted binding sites from the *fl3*, *sus1*, *c1*, and *vp1* promoters were synthesized and labeled with the Pierce Biotin 3′ End DNA Labeling Kit (Genewiz). Labeled single-stranded oligos were annealed to reverse complement unlabeled oligos. Probe sequences are listed in Supplementary Table 1.

Protein-DNA binding reactions included 50 ng of purified DED1-GST, 6 ng of biotin-labeled annealed oligonucleotides in 20 µL of 10 mM Tris (pH 7.5), 50 mM KCl, 1 mM DTT, 2.5% (v/v) glycerol, 5 mM $MgCl_2$, 50 µg/µL poly(dI-dC), and 0.05% (v/v) NP-40. The reactions were incubated at 25 °C for 20 min, electrophoresed on 6% (w/v) polyacrylamide gels, and transferred to Hybond N+ nylon membranes (Amersham Pharmacia Biotech). Biotin-labeled DNA was detected using a LightShift Chemiluminescent EMSA kit (Thermo Fisher Scientific) and imaged with an Amersham Imager 600.

## Histology
Histology analysis was performed as described[58]. Briefly, three 12 DAP self-pollinated ears segregating for *ded1-ref* in the W22 background were harvested. Individual kernels were selected, hand sectioned, and fixed 3.7% formaldehyde, 5% glacial acetic acid, and 50% ethanol overnight at 4 °C. Fixed tissues were embedded in paraffin and sectioned at 8 µm thickness. Normal embryo sections were stained with safranin and fast green, and all other sections were stained in basic fuchsin, washed, dried, and mounted. Imaging was completed with a Zeiss Axiophot light microscope and an Amscope digital camera.

## *Ded1* dosage series for gene expression analysis
W22 and *ded1-ref*/+ plants were self-pollinated and crossed reciprocally. At 11 DAP, endosperm tissue was dissected from W22 self-pollinated ears to sample the homozygous normal, DDD, genotype. Homozygous *ded1* mutant (ddd) endosperm was dissected from *ded1-ref*/+ self-pollinations. Heterozygous tissue was identified in reciprocal crosses by dissecting individual endosperm and genotyping corresponding embryos with a multiplex PCR using MYB73-F6-A, MYB73-R5 and LTR-F2 primers to amplify the normal and *ded1-ref* allele simultaneously. Independent ears were considered biological replicates, and endosperm RNA was extracted from a pool of five heterozygous kernels from each ear. The ddD dose derived from *ded1-ref*/+ plants crossed as female, and the DDd dose was from *ded1-ref*/+ crossed as male. Quantitative RT-PCR (qRT-PCR) was completed as described above.

## Reporting summary
Further information on research design is available in the Nature Research Reporting Summary linked to this article.

## Data availability
RNA-seq and DAP-seq data are available through the NCBI Accession GSE183304. The *ded1-ref* mutant is available at the Maize Genetics Cooperative Stock Center. Source data are provided with this paper.

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

## Acknowledgements

We thank L. Curt Hannah for critical evaluation of the paper as well as John Baier, Nicole S. Beisel, and Chi-Wah Tseung for technical assistance. This work was supported by the Vasil-Monsanto Endowment, National Science Foundation IOS awards 0404560 (to A.M.S.), 0606607 (to A.M.S.), 1623478 (to A.M.S.), and 1916804 (to A.G.) as well as National Institute of Food and Agriculture awards to A.M.S.: 2011-67003-30215 and 2018-51181-28419.

## Author contributions

D.D.: Methodology, Formal analysis, Investigation, Data Curation, Writing—Original Draft, Writing—Review & Editing, Visualization. J.M.: Methodology, Formal analysis, Investigation, Data Curation, Writing—Original Draft, Writing—Review & Editing, Visualization. M.G.: Methodology, Software, Formal analysis, Investigation, Visualization. S.N.C.: Methodology, Formal analysis, Investigation. R.D.: Investigation. X.Z.: Investigation. J.L.G.: Investigation. G.S.: Investigation. J.Z.: Investigation. W.B.B.: Methodology, Software, Investigation, Data Curation, Supervision. B.Y.: Methodology, Resources, Supervision. A.G.: Methodology, Resources, Supervision. A.M.S.: Conceptualization, Methodology, Software, Formal analysis, Investigation, Resources, Data Curation, Writing—Original Draft, Writing—Review & Editing, Visualization, Supervision, Project administration, Funding acquisition.

## Competing interests

The authors declare no competing interests.
