## [Peer Review File · Nature Communications]

Paternal imprinting of dosage-effect defective1 contributes to seed weight xenia in maizeREVIEWER COMMENTS

Reviewer #1 (Remarks to the Author):

This was a very interesting paper overall and provides a great case and possibly the first examples of an effect of a paternally imprinted gene, DED1, in seed development in maize, providing some evidence for the early hypothesized concept of xenia effects in seed development. The gene expression and mutant phenotypes also seem to support parental conflict theory which is not the case for all mutations in genes with parent-of-origin effects on gene expression.

The work in the mutant screen to identify mutations with dosage effects in seed development was well designed and the subsequent work to identify the gene as well as expand the network around it with a variety of genomic techniques helps to elucidate some of the mechanisms involved in the action of DED1.

For these reasons, I think this manuscript is a significant and valuable contribution to interactions between parental genomes during seed development in plants and is a valuable addition to the field.

I do have a number of specific suggestions/questions about the manuscript for clarity.

In reading the results, I originally had a question about the control for seed position within the ear but that was answered in the methods. Perhaps, mention it briefly in the results as well for clarity.

The conclusion is that the effect of mutant allele dosage on seed development depends on the differential expression of the maternal and paternal alleles. Was there any attempt to determine if there are any qualitative differences in the expression of the two alleles (splice site usage, transcription start site)? Admittedly, this would require SNPs in useful positions to resolve, but perhaps the strong mutant alleles would allow resolving this in reciprocal crosses. It is a minor point and if there is no evidence to distinguish this possibility it should at least be mentioned.

Regarding the use of the overlap between the DAP-seq and RNA-seq DEG gene lists to identify direct targets, I'd like to see a more rigorous analysis of the frequency of false positive targets considering the number of genes with binding that don't have expression differences. I worry that since there are so many genes bound by DED1 protein that are not targets based on lack of expression change, how many of the ones with expression change can be expected to be indirect targets despite DED1 being bound to the promoter (i.e. the binding of DED1 has no effect on gene expression in those cases and a downstream effect is actually responsible). Can the putative binding site help in this regard? Of course, this has its own caveats if DED1 is actually acting through another DNA-binding protein to bind to a gene that it then affects, but it would still be useful information for the reader with appropriate caveats.

Also, regarding up-regulation and down-regulation, there are of course possible feedback mechanisms that may also be involved and are unknown because a steady state RNA level in mutant cells are being examined, which are subject to many downstream effects. These effects may not be proximal to target binding by DED1.

It is very interesting to see that a maternally expressed gene, fl3, is under control of a paternally expressed gene and may provide some interesting insight into parental conflict.

In the discussion at the bottom of page 13 about the precocious differentiation events in the endosperm of ded1 mutants (i.e. DED1 normally delays these events as a PEG) is consistent with parental conflict theory.

In the discussion on page 14 line 314 it is mentioned that the homozygous plants are normal when it is presented earlier that they are smaller than wild type.

The cause of embryo arrest in *ded1* mutants is attributed to defects in the ESR or EAS. While these are both reasonable, there are also defects in the BETL which may be the critical defect with regard to embryo development. The issue is simply not completely resolved.

In the methods section about the RT-PCR experiments in lines 391-392, the use of random oligo dT primers is mentioned. It seems like this is a typo. Please specify if random oligos, oligo dT, or both were used.

In figure 3f, a row at the top showing the expression pattern of *Ded1* would be helpful to the reader for comparing its expression pattern with that of potential targets.

Reviewer #2 (Remarks to the Author):

In this manuscript, Dai et al. focus on *Ded1* function in maize endosperm development. I overall enjoyed reading the paper, which I find relatively thorough when it comes to *Ded1* function in regulating downstream targets and endosperm. I have quite some concerns nevertheless, as I find that there are some overstatements and claims not really supported by solid data.

The first one concerns the claim of the authors of *Ded1* being a "quantitative PEG", and I assume they use "quantitative" as an euphemism to say it is marginally imprinted. Indeed the results shown here suggests to me that its imprinting, if real, is really a minor phenomenon. First, Figure 1i shows that kernel weight resulting from crosses between the different *ded1* mutants is intermediate between parents. If there was a significant paternal effect of *ded1*, one would expect that the kernel weight would significantly look closer to the paternal kernel weight, and would not be an intermediate. Second, the authors claim that "the paternal allele in the reciprocal hybrids accounted for approximately $\frac{2}{3}$ of total transcript amplified" (l. 122-123). This claim seems based on a gel picture from Figure 2, which cannot be used to infer any quantitative value whatsoever, and which is rather inconclusive in general to me. Nevertheless the authors persist and base subsequent "dosage effect" claims (l. 255-265 and Figure 4) on this gel picture, by taking as a fact their extrapolation of parental transcript proportions from the gel. Finally, l. 129-138 is overclaiming the paternal effect on the loss of kernel weight and persists on the quantification of the paternal transcripts from the gel picture. The phenotypic effect is a loss of kernel weight of about 5% with paternal mutation while the homozygous mutants shows a reduction of about 90% compared to the normal parents. I think the numbers speak for themselves, i.e. the eventual paternal imprinting of *Ded1* plays a minor effect on the phenotype, which is mostly determined in a biparental manner.

In continuation with these, I found many statements to be overstatements across the manuscript. For example: l. 77-78 "A small fraction of *ded1* kernels are viable due to reduced expressivity". The authors write reduced expressivity as a fact, while it is merely their assumption. l. 217-218: "DED1-activated targets are associated with information processing functions". I don't see how the genes in this paragraph are involved in "information processing". l.290-292: *Ded1* is "an example where maternal inheritance reduces nutritional resources to a minimally sufficient level, while paternal inheritance increases resources to the progeny receiving functional *Ded1*." This is pure speculation, the only conclusion that can be done given the largely biparental determination of the kernel weight (see above) is instead additive effect of parental genome, and in any case doesn't tell anything about the evolutionary forces behind such as parental conflict. In the same line, l. 333-335, the results do not really support the kinship theory for the same reasons.

Other comment: l.78-79, the authors say that the homozygous plants are fully fertile, so I don't really

get why they work with heterozygous plants, which would be the strategy in case the homozygous plants would not produce seeds.

I have a problem with the following statements:

l. 88-92: "PCR of ded1-ref mutant cDNA amplified a 5' open reading frame (ORF) product but not a product 3' of the retrotransposon insertion suggesting ded1-ref expresses a transcript with a premature termination codon truncating the C-terminal acidic domain(Supplementary Fig. 2d)"

l. 95-99:"The ded1-1 to ded1-4 alleles cause premature termination codons that truncate the Ded1 ORF 5' of ded1-ref. Quantitative RT-PCR (qRT-PCR) analysis of 12 days after pollination (DAP) endosperm revealed decreased expression of ded1 in homozygous mutants for all alleles, which is consistent with the transcript being sensitive to nonsense mediated decay (Fig. 1h)."

According to the vocabulary used (codons, C terminal...), there seems to be a mix between mutations causing mRNA termination (ded1-ref) and those causing protein translation termination (the CRISPR mutants). Supplementary Fig. 2d shows no amplification of ded1-ref with primers 4 from cDNA, hence a truncated RNA, not a premature termination of protein translation.

In the same line, the authors conclude that "Quantitative RT-PCR (qRT-PCR) analysis of 12 days after pollination (DAP) endosperm revealed decreased expression of ded1 in homozygous mutants for all alleles, which is consistent with the transcript being sensitive to nonsense mediated decay". But the authors do not give information about where the primers used for qRT-PCR are located in ded1-ref, and if they're located after the TE insertion, then they give evidence of truncated RNA by their inability to amplify the cDNA, not a decreased expression...

l. 319-329: The authors' manuscript interestingly reaches similar findings to a previous work focusing on the MADS-box TF PHERES1 (Batista et al. 2019; PMID: 31789592) in Arabidopsis, but this reference appears nowhere in the manuscript. Since authors compare here their results to studies from other species, I think that discussing Batista et al paper is here really important.

Reviewer #3 (Remarks to the Author):

This manuscript by Dai et al., the authors mapped and validated the gene dosage-effect defective1 (ded1), which is a paternally expressed gene that regulates seed size and development. This gene encodes for a MYB family transcription factor. The authors utilized multiple alleles, including CRISPR alleles, to validate that this gene is responsible for seed development phenotype. Then, the authors did the follow-up studies of differential expression analysis and TF binding assays to better understand the consequences of this gene on the genome. I think this is a very well done study. I have a few minor comments that would help clarify some minor points in this already very nice manuscript.

Minor comments:

Lines 78-79. Please define "fully fertile" since it is not clear what that means in this context. From the images, it looks like plants have reduced seed set.

Line 86. What versions of the genome annotation were you comparing? There are now 5 numbered versions and the word "between" suggests a pairwise comparison.

Figure 1h. Please mark in 1f where the primers are used to make the plot in figure 1h. It is not clear currently.

Lines 123-125. What was the paternal bias in prior studies? It would be helpful to know if Ded1 is a

similarly quantitative PEG in past studies.

Point-by-point response to Reviewers

We thank the reviewers for their critical insights and helpful comments to improve our manuscript entitled, “Paternal imprinting of dosage-effect defective1 contributes to seed weight xenia in maize.” Our detailed responses to the individual comments follow.

Reviewer #1

Comment 1: In reading the results, I originally had a question about the control for seed position within the ear but that was answered in the methods. Perhaps, mention it briefly in the results as well for clarity.

Response: We edited the results to describe the ear sampling methods.

Comment 2: The conclusion is that the effect of mutant allele dosage on seed development depends on the differential expression of the maternal and paternal alleles. Was there any attempt to determine if there are any qualitative differences in the expression of the two alleles (splice site usage, transcription start site)? Admittedly, this would require SNPs in useful positions to resolve, but perhaps the strong mutant alleles would allow resolving this in reciprocal crosses. It is a minor point and if there is no evidence to distinguish this possibility it should at least be mentioned.

Response: We investigated three possible qualitative differences in *ded1* transcripts. First, are there qualitative differences in transcripts produced by maternal/paternal inheritance of normal alleles among maize varieties? Second, are there qualitative differences in the *ded1-ref* and Cas9-induced *ded1* alleles induced in cis- to the primary mutation sites reported in Figure 1 and Supplementary Figure 3? Third, do *ded1* mutants impact normal *Ded1* allele transcript levels?

As to the first option, we have not found convincing evidence of qualitatively different transcripts being produced by normal *Ded1* alleles. MaizeGDB has >30 annotated genomes, and only 2 genomes annotate alternative transcripts: B73 and Ms71. The Ms71 alternative transcript includes a long 5' UTR and truncated ORF. The MaizeGDB annotation notes report that only 33% of the annotated transcript is supported by mRNA-seq data, and this annotation partially overlaps with the experimentally-validated cDNA. In B73, the predicted alternate transcript has a cryptic 195-base intron in exon 1. Figure 1 shows endosperm RT-PCR designed to amplify 496 bp and 301 bp products for the two B73 annotated transcripts. The 301 bp splice variant from B73_v5 T01 does not appear to accumulate in either B73 or W22 inbred lines, while the 496 bp product from B73_v4 T01/B73_v5 T02 amplifies from both inbred lines. We added the following statement to the results, “No evidence was found for a predicted alternatively spliced transcript in the B73_v5 gene model, Zm00001eb050770.”

As to the second, 3' RACE of the *ded1-ref* transcript found the 3' end of the transcript to include 154 bases of the copia-like LTR transposon as the primary effect of this insertion mutation on the transcript. No additional qualitative effects were found, because the cDNA sequence 5' of the LTR insertion is identical to the W22 allele. Full length cDNA sequences of the CRISPR

alleles only had the polymorphisms reported in Supplementary Figure 3. No other evidence of qualitative differences was detected, and we did not modify the manuscript to emphasize the lack of secondary cis- effects.

As to the third, we designed qRT-PCR primers to amplify the normal *Ded1* allele in the dosage series shown in Figure 4a. Paternal inheritance of the normal allele (dd/D) had 50% expression level of the homozygous *Ded1* genotype (DDD). Maternal inheritance of the normal allele (DD/d) had 15% expression of the homozygous normal genotype. Expression of the normal *Ded1* allele was not detected in the homozygous mutant. Considering the variance of biological sampling and qRT-PCR, these data are consistent with the PEG pattern of expression observed in Figure 2b and in Zhang et al (2011) and Waters et al (2011). We have updated Figure 4a to include this experiment. Note that we found a calculation error in the qRT-PCR analysis in Figure 4a. The original submission stated a sample size of 9. The sampling was 3 biological replicate RNA extractions from different ears with 3 technical replicates per biological replicate. The graphs now show individual points for the average values of the three biological replicates. We also repeated qRT-PCR analysis of *de18* and *se1* because these genes had high variance among technical replicates in the initial submission. Revised ANOVA confirmed all DEGs showed differential expression. Although *se1* transcript levels were significantly affected in the ANOVA ($p=0.04$), the Tukey's HSD test no longer showed significant groups in the dosage series, and this gene was removed. We edited the discussion to be clear that only a subset of DED1 downstream genes were observed to have dose-dependent responses in transcript levels.

Comment 3: Regarding the use of the overlap between the DAP-seq and RNA-seq DEG gene lists to identify direct targets, I'd like to see a more rigorous analysis of the frequency of false positive targets considering the number of genes with binding that don't have expression differences. I worry that since there are so many genes bound by DED1 protein that are not targets based on lack of expression change, how many of the ones with expression change can be expected to be indirect targets despite DED1 being bound to the promoter (i.e. the binding of DED1 has no effect on gene expression in those cases and a downstream effect is actually responsible). Can the putative binding site help in this regard? Of course, this has its own caveats if DED1 is actually acting through another DNA-binding protein to bind to a gene that it then affects, but it would still be useful information for the reader with appropriate caveats.

Response: We reviewed our analysis of the DAP-seq and DEG overlap and revised the results text, Figure 3c, and Supplementary Figure 6d to reflect this analysis more accurately. Please note that in contrast to ChIP-seq, DAP-seq reports all possible transcription factor binding sites without influence of chromatin state or other transcription factors in the developmental context. In our original submission, we reported 20,381 genes had a DED1 DAP-seq peak -1 kbp to +1 kbp of the annotated transcription unit. We double-checked the analysis of the DAP-seq peaks and found there were only 15,367 genes that met these criteria. For the 5,860 genes with DAP-seq peaks that are -1 kbp to +100 bp of transcriptional start sites, the *ded1-ref* and normal sibling endosperm RNA-seq analysis only detected expression for only 2,762 of these genes. Thus, the analysis detected 438 direct targets from 2,762 tests of genes bound by DED1. This is a significant enrichment beyond random overlap with a cumulative hypergeometric probability of this many DEGs being bound by DED1 through chance at 2.2×10^{-8} .

As a reality check, we compared our results to other ChIP-seq and DAP-seq studies in maize, Arabidopsis, and *Pseudomonas aeruginosa* (Li et al, 2016; Batista et al, 2019; Huang et al, 2019; Dong et al, 2020). These studies used filtered binding sites overlapped with DEGs from mutant-normal RNA-seq comparisons to identify direct targets for 17 transcription factors. On

average, 11% of DEGs were identified as direct targets, while we found 23% of DEGs were direct targets of DED1. In the plant studies, 7%, 9%, and 16% of filtered binding sites had a DEG in a mutant-normal mRNA-seq comparison. For DED1, 18% of the filtered binding sites have a DEG. Based on these comparisons, the results reported for DED1 are in-line with the binding site/DEG overlaps observed in prior publications.

Li et al. (2016) <https://doi.org/10.1073/pnas.1602960113>

Batisita et al. (2019) <https://doi.org/10.7554/elife.50541>

Huang et al (2019) <https://doi.org/10.1038/s41467-019-10778-w>

Dong et al. (2020) <https://doi.org/10.1073/pnas.2005014117>

Comment 4: Also, regarding up-regulation and down-regulation, there are of course possible feedback mechanisms that may also be involved and are unknown because a steady state RNA level in mutant cells are being examined, which are subject to many downstream effects. These effects may not be proximal to target binding by DED1.

Response: We modified the Discussion with the following paragraph:

“At a molecular level, DED1 is a transcription factor that regulates early endosperm gene expression in a dosage sensitive manner. Consequently, the study has limitations in identifying DEGs and direct target genes. DEGs were identified from 12 DAP endosperm tissues that compared homozygous mutant to a pool of normal tissues with three doses of the normal *Ded1* allele. These mixed endosperm genotypes in the normal RNA extraction as well as sampling after 6 DAP, when *Ded1* is maximally expressed, likely reduced the statistical power to detect DEGs. Moreover, steady-state transcript levels combine both direct transcriptional responses to DED1 protein and feedback mechanisms that cause indirect transcript level changes.”

Comment 5: In the discussion at the bottom of page 13 about the precocious differentiation events in the endosperm of *ded1* mutants (i.e. DED1 normally delays these events as a PEG) is consistent with parental conflict theory.

Response: We added a statement connecting Haig & Westoby’s (1989) parental conflict theory prediction of PEGs that promote endosperm growth with *ded1* function.

Comment 6: In the discussion on page 14 line 314 it is mentioned that the homozygous plants are normal when it is presented earlier that they are smaller than wild type.

Response: We modified the language to be consistent. We meant *ded1-ref* mutants can be viable, relatively healthy, and fertile. We did not mean that the homozygous mutant plants are completely equivalent to isogenic *Ded1* normal plants.

Comment 7: The cause of embryo arrest in *ded1* mutants is attributed to defects in the ESR or EAS. While these are both reasonable, there are also defects in the BETL which may be the critical defect with regard to embryo development. The issue is simply not completely resolved.

Response: We expanded the discussion of the mutant embryo arrest to include considerations of BETL defects. We believe the bulk of studies favor a more direct role in nutrient transport through the ESR and/or EAS. Mutations that cause specific BETL defects do not impact embryo development arguing that the BETL is generally not essential for embryo development (Costa et al., 2012; Sosso et al., 2015). Indirect effects on BETL development and function have been observed in many defective kernel mutants and appears to be a developmental response when overall seed sink strength is reduced (discussed more in Fouquet et al., 2004). In the case of DED1, nutrient transporters in the ESR and EAS are direct targets, while related transporters in the BETL are indirectly affected in the mutant.

Comment 8: In the methods section about the RT-PCR experiments in lines 391-392, the use of random oligo dT primers is mentioned. It seems like this is a typo. Please specify if random oligos, oligo dT, or both were used.

Response: We edited the methods to remove the typographical error. The reverse transcriptase reactions were primed with oligo(dT) primer from Promega.

Comment 9: In figure 3f, a row at the top showing the expression pattern of *Ded1* would be helpful to the reader for comparing its expression pattern with that of potential targets.

Response: We added the *Ded1* expression pattern to Figure 3f and removed Supplementary Figure 4c. The results text was reorganized so that the endosperm cell-specific expression pattern of *Ded1* is explained in conjunction with Figure 3f.

Reviewer #2

Comment 1: The first one concerns the claim of the authors of *Ded1* being a "quantitative PEG", and I assume they use "quantitative" as an euphemism to say it is marginally imprinted. Indeed the results shown here suggests to me that its imprinting, if real, is really a minor phenomenon.

Response: We revised the results section to provide a more detailed description of prior studies that identified *ded1* as a PEG. We also added qRT-PCR results for *Ded1* normal allele expression levels for the *ded1-ref* gene dosage series in Figure 4a. These additional data estimate that paternal inheritance of *Ded1* contributes 50% of the transcript, while maternal inheritance contributes 15% of the transcript. Plant-to-plant biological variance or feedback mechanisms may account for the non-additive transcript levels observed in the dosage series. However, these results agree with prior mRNA-seq studies and Figure 2b.

We use the term quantitative PEG to avoid confusion with a common misperception that plant imprinted genes are exclusively expressed in a parent-of-origin specific manner throughout the entirety of endosperm development. In proposing kinship theory, Haig and Westoby (1989) suggested that parent specific gene expression would eventually be selected for exclusive expression from one parent's allele. However, they also recognized that forms of balancing selection between loci acting in endosperm development pathways might limit parent specific gene expression to be "more strongly expressed" from one parent. The maize *Meg1* gene provides a good example of functional angiosperm imprinting. *Meg1* is expressed from the maternal allele from 4-6 DAP, the gene shows quantitative maternal-biased expression at 10 DAP and is biparental by 15 DAP (Gutiérrez-Marcos et al., 2004). Nevertheless, imprinted, maternal expression of *Meg1* limits the extent of BETL cell specification and differentiation to limit resources to individual progeny (Costa et al., 2012).

For *ded1*, there is significant evidence from other laboratory groups that this locus is an imprinted PEG based on mRNA-seq. Waters et al. (2011) used 14 DAP endosperm and found *ded1* to be a PEG based on Chi-square statistics. In B73xMo17 crosses, *ded1* ranked 75th most significant p-value of the 12,571 genes analyzed. In Mo17xB73 crosses, *ded1* ranked 52nd most significant p-value for the 12,571 genes. Zhang et al (2011) used 10 DAP endosperm, and *ded1* was found to be among their 111 high-confidence PEGs that had at least a 5-fold excess in expression from the paternal allele from 11,370 genes analyzed. Zhang et al (2014) also found *ded1* to be among “high quality imprinted transcripts” that included 79 PEGs identified in their 12 DAP reciprocal crosses. In these prior studies, *ded1* is among the highest confidence PEGs identified. Our data in Figures 2b and 4a confirm these prior observations in both a wild-type and heterozygous mutant context. We hope the addition of the *Ded1* normal allele qRT-PCR data and more detailed explanation of prior studies convinces the reviewer that there are multiple lines of independent evidence that *Ded1* is a PEG in maize.

Gutiérrez-Marcos et al. (2004) <https://doi.org/10.1105/tpc.019778>

Costa et al. (2012) <https://doi.org/10.1016/j.cub.2011.11.059>

Waters et al. (2011) <https://doi.org/10.1105/tpc.111.092668>

Zhang et al (2011) <https://doi.org/10.1073/pnas.1112186108>

Zhang et al (2014) <https://doi.org/10.1101/gr.155879.113>

Comment 2: Figure 1i shows that kernel weight resulting from crosses between the different *ded1* mutants is intermediate between parents. If there was a significant paternal effect of *ded1*, one would expect that the kernel weight would significantly look closer to the paternal kernel weight, and would not be an intermediate.

Response: All of the kernels in Figure 1i have severe developmental defects. Even with large sample sizes, the mutant kernel weights do not have sufficient statistical power to distinguish heteroallelic combinations. Due to this lack of statistical power, we limited our statements in the results to the trend that all heteroallelic combinations are intermediate weight relative to the strong and weak allele homozygous phenotypes.

Comment 3: The authors claim that “the paternal allele in the reciprocal hybrids accounted for approximately $\frac{2}{3}$ of total transcript amplified” (l. 122-123). This claim seems based on a gel picture from Figure 2, which cannot be used to infer any quantitative value whatsoever, and which is rather inconclusive in general to me. Nevertheless the authors persist and base subsequent “dosage effect” claims (l. 255-265 and Figure 4) on this gel picture, by taking as a fact their extrapolation of parental transcript proportions from the gel.

Response: We edited the results to provide more detailed explanation of prior mRNA-seq studies to detect maize imprinted genes as well as adding supporting qRT-PCR data to Fig. 4a. Briefly, Zhang et al (2011) found that the paternal allele of *ded1* accounts for 75-77% of total transcript at 10 DAP. Zhang et al (2014) did not provide maternal and paternal expression ratios for 12 DAP endosperm. Waters et al. (2011) found 53-68% of the total transcript at 14 DAP was from the paternal allele. Our estimate from the gel shown in Figure 2b was further tested using qRT-PCR in the *ded1-ref* dosage series shown in Figure 4a. These qRT-PCR data indicate 50% paternal and 15% maternal expression of *Ded1*.

Comment 4: Line 129-138 is overclaiming the paternal effect on the loss of kernel weight and persists on the quantification of the paternal transcripts from the gel picture. The phenotypic effect is a loss of kernel weight of about 5% with paternal mutation while the homozygous mutants shows a reduction of about 90% compared to the normal parents. I think the numbers speak for themselves, i.e. the eventual paternal imprinting of *Ded1* plays a minor effect on the phenotype, which is mostly determined in a biparental manner.

Response: We modified this section to explain that *ded1* parent-of-origin expression levels were inferred from published mRNA-seq experiments as well as RT-PCR results shown in Figure 2b. The language in the results was modified to claim correlation with *Ded1* expression level rather than causation. The Figure 2 legend was modified to change “determines” to “affects.” No claims are made about whether the paternal inheritance of *ded1* mutant alleles cause a “minor” or “major” seed weight effect. We only claim a statistically significant reduction in seed weight correlated with parent-of-origin inheritance.

Comment 5: Line 77-78 "A small fraction of *ded1* kernels are viable due to reduced expressivity". The authors write reduced expressivity as a fact, while it is merely their assumption.

Response: We deleted “due to reduced expressivity.”

Comment 6: Line 217-218: "DED1-activated targets are associated with information processing functions". I don't see how the genes in this paragraph are involved in "information processing".

Response: We modified “information processing” to “endosperm developmental processes.”

Comment 7: Line 290-292: *Ded1* is "an example where maternal inheritance reduces nutritional resources to a minimally sufficient level, while paternal inheritance increases resources to the progeny receiving functional *Ded1*." This is pure speculation, the only conclusion that can be done given the largely biparental determination of the kernel weight (see above) is instead additive effect of parental genome, and in any case doesn't tell anything about the evolutionary forces behind such as parental conflict. In the same line, l. 333-335, the results do not really support the kinship theory for the same reasons.

Response: We modified the Line 290-292 statement to remove speculative claims on minimal sufficiency. It now reads, “The *Ded1* parent-of-origin expression pattern and seed weight outcomes illustrate an example where maternal inheritance confers lower nutritional resource uptake and accumulation of seed reserves in progeny, while paternal inheritance of functional *Ded1* increases nutritional resource uptake and seed reserve accumulation.”

Regarding the final statement in the discussion, Haig and Westoby's (1989) paper provides the detailed evolutionary argument for the parent specific gene expression (PSGE) hypothesis. This hypothesis is now referred to as the parental conflict hypothesis or kinship theory. In their paper, they specifically state:

“The major prediction of the hypothesis is that there exists a class of loci for which paternally derived alleles are considerably more strongly expressed than maternally derived alleles. These loci are predicted to encode proteins responsible for acquiring resources from the mother for the offspring, and the

PSGE in question should be found in the offspring tissue that acquires the resources.”

Four studies (including this manuscript) using three different experimental methods found paternal bias in expression of *ded1*. These data satisfy the first part of Haig and Westoby's prediction. A subset of DED1 target genes is known to function in nutrient acquisition in the endosperm, and *ded1* is expressed specifically in the endosperm. These molecular characteristics satisfy the second part of Haig and Westoby's prediction. We respectfully disagree with the reviewer and have left the final statement of the discussion unchanged.

Comment 8: Line78-79, the authors say that the homozygous plants are fully fertile, so I don't really get why they work with heterozygous plants, which would be the strategy in case the homozygous plants would not produce seeds.

Response: We had two reasons that we did not use homozygous plants for most assays in the study. First, viable homozygous *ded1-ref* kernels are at very low frequency. Practically, we could not incorporate mutant kernels into field experiments and reliably obtain sufficient mutant plants. Second, mutant plants are slightly smaller than heterozygous normal siblings. Kernels developed on homozygous mutant ears could have phenotypes caused by the maternal parent phenotype instead of the kernel genotype. By comparing kernels of different genotypes on the same ear, we control for both environmental variation and the influence of the maternal plant's health.

Comment 9: I have a problem with the following statements:

Line 88-92: "PCR of *ded1-ref* mutant cDNA amplified a 5' open reading frame (ORF) product but not a product 3' of the retrotransposon insertion suggesting *ded1-ref* expresses a transcript with a premature termination codon truncating the C-terminal acidic domain(Supplementary Fig. 2d)"

Line 95-99:"The *ded1-1* to *ded1-4* alleles cause premature termination codons that truncate the Ded1 ORF 5' of *ded1-ref*. Quantitative RT-PCR (qRT-PCR) analysis of 12 days after pollination (DAP) endosperm revealed decreased expression of *ded1* in homozygous mutants for all alleles, which is consistent with the transcript being sensitive to nonsense mediated decay (Fig. 1h)."

According to the vocabulary used (codons, C terminal...), there seems to be a mix between mutations causing mRNA termination (*ded1-ref*) and those causing protein translation termination (the CRISPR mutants). Supplementary Fig. 2d shows no amplification of *ded1-ref* with primers 4 from cDNA, hence a truncated RNA, not a premature termination of protein translation.

Response: We modified the statement in lines 88-92 to describe the *ded1-ref* transcript more accurately. The revised text reads: "PCR of *ded1-ref* mutant cDNA amplified a 5' open reading frame (ORF) product but not a product 3' of the retrotransposon insertion (Supplementary Fig. 2d). The full-length cDNA sequence of the *ded1-ref* transcript ORF includes part of the retrotransposon sequence and a predicted protein lacking part of the C-terminal acidic domain (Fig. 1f)."

Comment 10: In the same line, the authors conclude that "Quantitative RT-PCR (qRT-PCR) analysis of 12 days after pollination (DAP) endosperm revealed decreased expression of

ded1 in homozygous mutants for all alleles, which is consistent with the transcript being sensitive to nonsense mediated decay". But the authors do not give information about where the primers used for qRT-PCR are located in ded1-ref, and if they're located after the TE insertion, then they give evidence of truncated RNA by their inability to amplify the cDNA, not a decreased expression...

Response: We have updated Figure 1f to indicate the positions of the qRT-PCR products used for Figures 1h, 2a, and 4a. We also modified the results text to emphasize that the qRT-PCR product in 1h and 2a is from exon 1 of the gene.

Comment 11: Line. 319-329: The authors' manuscript interestingly reaches similar findings to a previous work focusing on the MADS-box TF PHERES1 (Batista et al. 2019; PMID: 31789592) in Arabidopsis, but this reference appears nowhere in the manuscript. Since authors compare here their results to studies from other species, I think that discussing Batista et al paper is here really important.

Response: We added the following paragraph to the discussion:

"No Arabidopsis PEGs have been identified that impact seed development in diploid crosses. However, several PEG loci function in blocking 2n x 4n crosses⁵⁹. In addition, the redundant MADS-box transcription factor loci, *phe1* and *phe2*, can also rescue 2n x 4n seed development⁶⁰. Although *phe2* is biparentally expressed, *phe1* is a PEG, and the double mutant suppresses interploidy seed abortion when inherited through pollen. PHE1 target genes have overlapping developmental functions with DED1 targets such as a bias for regulating PEGs, epigenetic regulators, auxin biosynthesis genes, and non-imprinted transcription factors. Potentially, other PEGs within Arabidopsis have xenia functions analogous to *ded1*."

Reviewer #3

Comment 1: Lines 78-79. Please define "fully fertile" since it is not clear what that means in this context. From the images, it looks like plants have reduced seed set.

Response: We changed "fully fertile" to "fertile." Fully was meant to emphasize that viable male and female gametophytes were produced.

Comment 2: Line 86. What versions of the genome annotation were you comparing? There are now 5 numbered versions and the word "between" suggests a pairwise comparison.

Response: When we first identified the *ded1-ref* allele, B73_v2 and B73_v3 had incomplete gene models. By the time we were writing the manuscript, B73_v4 and both B73_v5 annotation versions "e" and "eb" had corrected the gene model to our experimentally validated cDNA sequence. However, B73_v5 annotates transcript T01 with alternative splicing of the first exon of *ded1*. We could not validate this shorter transcript form. We modified the statement to: "The predicted mRNA sequence for *ZmMyb73* varies among B73 genome annotations from version 2 to version 5."

Comment 3: Figure 1h. Please mark in 1f where the primers are used to make the plot in figure 1h. It is not clear currently.

Response: We have updated Figure 1f to show the qRT-PCR products for Figures 1h, 2a, and 4a.

Comment 4: Lines 123-125. What was the paternal bias in prior studies? It would be helpful to know if *Ded1* is a similarly quantitative PEG in past studies.

Response: We modified the results section for Figure 2b and added a panel to Figure 4a. Briefly, Zhang et al (2011) found that the paternal allele of *ded1* accounts for 75-77% of total transcript at 10 DAP. Zhang et al (2014) did not provide maternal and paternal expression ratios for 12 DAP endosperm. Waters et al. (2011) found 53-68% of the total transcript at 14 DAP was from the paternal allele. Our estimates from 13 DAP endosperm are between these observations. In Figure 4a, we include the relative level of *Ded1* normal allele in the *ded1-ref* dosage series. This shows the paternal allele contributes 50% of homozygous normal levels, while the maternal alleles contribute 15% of homozygous normal levels.

REVIEWERS' COMMENTS

Reviewer #1 (Remarks to the Author):

After reading through the author's responses to my comments and the comments of the other reviewers, I am satisfied with the manuscript as it now stands.

I would just like to clarify my earlier comment on how many of the genes are true DED1 targets (reviewer 1 comment 3 with response in the author's rebuttal letter). It is great that the statistical support for this conclusion is included in the manuscript now. My original intent was not that the gene list was not enriched for targets, but rather that it is unlikely that the list is made up 100% of targets with no false positives. I was merely looking for an estimate of what percentage of these genes may be false positives based on the frequency of non-bound genes with transcriptional changes and DED1-bound genes with no transcriptional change. It is not an essential point but would be useful for subsequent researchers interested in following up on DED1-downstream genes.

Reviewer #2 (Remarks to the Author):

This is a revised version of the manuscript by Dai et al. The version is definitely improved in my opinion.

This being said, I still do have some concerns, most of them rather minor.

Probably the biggest issue still for me is the claim of a xenia effect. Surely, I agree that *ded1* fits the idea in principle of a paternal gene having an effect on seed size, and the authors indeed do not claim a "major" effect. My previous concern is still here nevertheless: can we really talk about a xenia effect when a paternal inheritance of the mutant allele decrease kernel size of about 5% while biparental inheritance of the mutation leads to a 90% decrease in size? So I'm not denying the result of a paternal effect, rather its significance/extent (which again I know the authors do not claim, but by claiming a xenia effect also in the title of the manuscript, I imagine it's still an indirect statement). I suggest in the discussion some sentences to nuance the claim of a xenia effect by commenting on the extent of the phenotypic effect of the paternal mutation in *ded1*. Perhaps one way is to comment that most likely, the paternal influence on seed size has a multigenic basis with many genes of small effect, as illustrated by the absence of phenotypic defects in Arabidopsis PEG mutants (Wolff et al 2015), and I believe this genetic basis is also in accordance with the parental conflict model. Or alternatively/in complement, the authors may explain the stronger effect of the biparental *ded1* mutation compared to the paternal mutation by a likely compensation by the maternal allele that takes place in the absence of a functional paternal allele.

l. 73. "A small fraction of *ded1* kernels is viable". I didn't notice this in the first version, my bad, but it would be informative to have numbers on what "a small fraction" means, by including some germination rate or so, to have an idea of the severity of the *ded1* mutation. Is it 1%? 10%? more?

l. 75. "Self-pollination of *ded1*-ref plants produces all mutant seeds with accelerated anthocyanin accumulation in the aleurone" is an overstatement to me, unless the authors measured anthocyanin content. Plus, if we assume purple = anthocyanins, then it seems that the normal seeds accumulate more than the *ded1* seeds to me according to Fig. 1d and 1e. Is it a label problem?

Discussion regarding the parental conflict: I actually didn't disagree with the authors that *ded1* matches the expectation of the parental conflict theory; l. 285-286 the authors wrote that "*ded1* provides a critical example in support of parental conflict theory", which is what I disagree on, as it is not the same as matching the expectations of the theory. That might seem like semantic, and yet because this theory is still stirring heated discussions, I imagine it is important to be accurate on

claims if we want to have a constructive discussion about it. Therefore, I do agree with the authors writing in l. 317 ("A PEG like Ded1 that promotes endosperm growth is a direct prediction of parental conflict theory as proposed by Haig and Westoby"), but I do not agree with l. 285-286. Please rephrase l. 285-286 in a similar frame as l.317.

In the same line, the authors wrote l.289-291, "where maternal inheritance confers lower nutritional resource uptake and accumulation of seed reserves in progeny,...". This claim is not correct to me: the maternal inheritance has limited/no effect on seed reserve, while the paternal inheritance promotes it. Now it sounds like the maternal inheritance inhibits reserve accumulation. Please reformulate.

Point-by-point response to Reviewers

We thank the reviewers for reevaluating our manuscript entitled, “Paternal imprinting of *dosage-effect defective1* contributes to seed weight xenia in maize.” Our detailed responses to the individual comments follow.

Reviewer #1

Comment 1: I would just like to clarify my earlier comment on how many of the genes are true DED1 targets (reviewer 1 comment 3 with response in the author’s rebuttal letter). It is great that the statistical support for this conclusion is included in the manuscript now. My original intent was not that the gene list was not enriched for targets, but rather that it is unlikely that the list is made up 100% of targets with no false positives. I was merely looking for an estimate of what percentage of these genes may be false positives based on the frequency of non-bound genes with transcriptional changes and DED1-bound genes with no transcriptional change. It is not an essential point but would be useful for subsequent researchers interested in following up on DED1-downstream genes.

Response: We agree with the reviewer that genome-wide analyses will contain false positive identifications due to the large number of statistical tests completed. However, we do not think comparing the two frequencies requested can give an estimate of false positive DED1 targets based on the following rationale.

First, “non-bound” DEGs may have true positive, regulatory DED1 binding sites that did not meet the criterion that the binding site must be within -1 kbp to +100 bp of the transcriptional start site (TSS). Regulatory sequences can be hundreds of kbp removed from the TSS. Classifying “non-bound genes” based on a narrow criterion for a promoter falsely categorizes genes that have regulatory DED1-binding sites outside the promoter window.

Second, standard DEG analysis has a limited power to detect genes that lack transcriptional change. Differential gene expression analysis algorithms, including DEseq2, only detect 50-70% of true positive DEGs in simulated data sets (Love et al, 2014). Calling genes in our data set that did not meet FDR statistics and the 4-fold differential expression criteria for a DEG as “no transcriptional change” is likely to categorize 5-15% of these genes as “not DEG” when they are actually differentially expressed.

We have not been able to find an example in the literature to estimate a frequency of false positive DED1 targets solely with the data reported in the manuscript. False positives have been estimated using meta-analysis of genome-wide studies. For example, Fisher (2017) examined p53 targets and found about 50-70% of p53 targets reported in a given study come from 116 higher confidence targets. Each of these higher confidence targets are reported in at least six of sixteen independent investigations. We used standard methods for genome-wide target identification with conservative criteria for promoters and DEGs to avoid false positives. We infer our analysis is likely to be at least comparable in statistical confidence to prior studies.

Love et al (2014) <https://genomebiology.biomedcentral.com/articles/10.1186/s13059-014-0550-8>

Fisher (2017) <https://www.nature.com/articles/onc2016502>

Reviewer #2

Comment 1: Probably the biggest issue still for me is the claim of a xenia effect. Surely, I agree that *ded1* fits the idea in principle of a paternal gene having an effect on seed size, and the authors indeed do not claim a “major” effect. My previous concern is still here

nevertheless: can we really talk about a xenia effect when a paternal inheritance of the mutant allele decrease kernel size of about 5% while biparental inheritance of the mutation leads to a 90% decrease in size? So I'm not denying the result of a paternal effect, rather its significance/extent (which again I know the authors do not claim, but by claiming a xenia effect also in the title of the manuscript, I imagine it's still an indirect statement). I suggest in the discussion some sentences to nuance the claim of a xenia effect by commenting on the extent of the phenotypic effect of the paternal mutation in *ded1*. Perhaps one way is to comment that most likely, the paternal influence on seed size has a multigenic basis with many genes of small effect, as illustrated by the absence of phenotypic defects in Arabidopsis PEG mutants (Wolff et al 2015), and I believe this genetic basis is also in accordance with the parental conflict model. Or alternatively/in complement, the authors may explain the stronger effect of the biparental *ded1* mutation compared to the paternal mutation by a likely compensation by the maternal allele that takes place in the absence of a functional paternal allele.

Response: Although we do not make a claim in the manuscript of a major paternal effect on seed weight, a 5-10% grain-weight genetic effect is considered a large effect locus in maize quantitative genetics and breeding. For example, *Nature Communications* recently published a major effect QTL that alters maize grain weight by 13% between the heavy and light alleles (Ning et al., 2021).

We agree with the reviewer that the contrast between the paternal and homozygous *ded1* phenotypes should be clearly stated early in the manuscript. We edited the abstract to emphasize that there are both paternal and homozygous mutant phenotypes for *ded1*. It now reads: "Hypomorphic alleles show a 5-10% seed weight reduction when *ded1* is transmitted through the male, while homozygous mutants are defective with a 70-90% seed weight reduction."

We would prefer not to speculate on the number of paternal effect genes that contribute to seed size in maize. Our genetic screen was not to saturation and did not screen for paternal-effects that fail to show a defective kernel phenotype in the homozygous state.

The *ded1-ref* dosage series in Fig 4a argues against maternal allele compensation when *ded1* is paternally inherited. Only 15% of normal *Ded1* levels were observed in 12 DAP endosperm. This suggests a 2-fold reduction relative to normal levels of maternal expression estimated by prior RNA-seq studies and the RT-PCR CAPS marker (Fig. 2b).

Ning et al. (2021) <https://www.nature.com/articles/s41467-021-26123-z>

Comment 2: l. 73. "A small fraction of *ded1* kernels is viable". I didn't notice this in the first version, my bad, but it would be informative to have numbers on what "a small fraction" means, by including some germination rate or so, to have an idea of the severity of the *ded1* mutation. Is it 1%? 10%? more?

Response: We observe a variable frequency of viable mutant kernels from ear to ear. When mutants display an unpredictable range in phenotype this is referred to as variable expressivity. Expressivity contrasts to reduced penetrance in which a fraction of homozygous mutants would be indistinguishable from normal genotypes. We modified the statement as follows. "Ears segregating for *ded1* homozygotes show variable expressivity with 0-15% of mutant kernels being viable. Viable seeds develop into slightly smaller, fertile plants (Fig. 1c and Supplementary Fig. 1d)."

Comment 3: l. 75. "Self-pollination of *ded1-ref* plants produces all mutant seeds with accelerated anthocyanin accumulation in the aleurone" is an overstatement to me, unless the authors measured anthocyanin content. Plus, if we assume purple = anthocyanins, then it seems that the normal seeds accumulate more than the *ded1* seeds to me according to Fig. 1d and 1e. Is it a label problem?

Response: Fig. 1d shows a normal *Ded1* sibling at 19 DAP. At this stage anthocyanins are beginning to accumulate in the aleurone and a mix of yellow and light purple tissues are visible through the pericarp. Normal kernels only become completely purple, as seen in the W22 panel of Fig. 1a, by 30 DAP. Fig. 1e shows homozygous *ded1-ref* kernels at 19 DAP. Differential growth of the maternal pericarp and seed causes the aleurone to become separated from the pericarp by this stage of development. The white/refractive tissue in the photograph is the separated pericarp. The sections in Fig. 1b show that the *ded1* aleurone makes contact with the pericarp in the center of the kernel crown. This contact is needed to allow light to reflect from the aleurone through the pericarp. The only seed color visible in kernel crown region of Fig. 1e is an intense, dark purple. This intense purple color does not develop in normal W22 kernels. We hope our reported observations, the photographic evidence of a more intense purple color accumulating 11 days before normal, and the up-regulation of an anthocyanin biosynthetic gene in *ded1-ref* mutants (Fig. 4a) are sufficient to support the claim.

Comment 4: Discussion regarding the parental conflict: I actually didn't disagree with the authors that *ded1* matches the expectation of the parental conflict theory; l. 285-286 the authors wrote that "*ded1* provides a critical example in support of parental conflict theory", which is what I disagree on, as it is not the same as matching the expectations of the theory. That might seem like semantic, and yet because this theory is still stirring heated discussions, I imagine it is important to be accurate on claims if we want to have a constructive discussion about it. Therefore, I do agree with the authors writing in l. 317 ("A PEG like *Ded1* that promotes endosperm growth is a direct prediction of parental conflict theory as proposed by Haig and Westoby"), but I do not agree with l. 285-286. Please rephrase l. 285-286 in a similar frame as l.317.

Response: We modified the sentence to: "Thus, *ded1* provides a critical example in relationship to parental conflict theory."

Comment 5: In the same line, the authors wrote l.289-291, "where maternal inheritance confers lower nutritional resource uptake and accumulation of seed reserves in progeny,...". This claim is not correct to me: the maternal inheritance has limited/no effect on seed reserve, while the paternal inheritance promotes it. Now it sounds like the maternal inheritance inhibits reserve accumulation. Please reformulate.

Response: We deleted: "maternal inheritance confers lower nutritional resource uptake and accumulation of seed reserves in progeny, while."

The statement now reads: "The *Ded1* parent-of-origin expression pattern and seed weight outcomes illustrate an example where paternal inheritance of functional *Ded1* increases nutritional resource uptake and seed reserve accumulation."